# Optimal Aggregation of Prediction Intervals under Unsupervised Domain Shift

**Jiawei Ge**[*]
Operations Research & Financial Engineering
Princeton University
jg5300@princeton.edu

**Debarghya Mukherjee**[*]
Department of Mathematics and Statistics
Boston University
mdeb@bu.edu

**Jianqing Fan**
Operations Research & Financial Engineering
Princeton University
jqfan@princeton.edu

## Abstract

As machine learning models are increasingly deployed in dynamic environments, it becomes paramount to assess and quantify uncertainties associated with distribution shifts. A distribution shift occurs when the underlying data-generating process changes, leading to a deviation in the model's performance. The prediction interval, which captures the range of likely outcomes for a given prediction, serves as a crucial tool for characterizing uncertainties induced by their underlying distribution. In this paper, we propose methodologies for aggregating prediction intervals to obtain one with minimal width and adequate coverage on the target domain under unsupervised domain shift, under which we have labeled samples from a related source domain and unlabeled covariates from the target domain. Our analysis encompasses scenarios where the source and the target domain are related via i) a bounded density ratio, and ii) a measure-preserving transformation. Our proposed methodologies are computationally efficient and easy to implement. Beyond illustrating the performance of our method through real-world datasets, we also delve into the theoretical details. This includes establishing rigorous theoretical guarantees, coupled with finite sample bounds, regarding the coverage and width of our prediction intervals. Our approach excels in practical applications and is underpinned by a solid theoretical framework, ensuring its reliability and effectiveness across diverse contexts.

## 1 Introduction

In the modern era of big data and complex machine learning models, extensive data collected from diverse sources are often used to build a predictive model. However, the assumption of independent and identically distributed (i.i.d.) data is frequently violated in practical scenarios. Take algorithmic fairness as an example: historical data often exhibit sampling biases towards certain groups, like females being underrepresented in credit card data. Over time, the differences in group proportions have diminished, leading to distribution shifts. Consequently, models trained on historical data may face shifted distributions during testing, and proper adjustments are needed. Distribution shift has garnered significant attention from statistical and machine learning communities under various names, i.e., transfer learning [PY09, WKW16], domain adaptation [FVRA21], domain generalization [ZLQ+22, WLL+22], continual learning [DLAM+21, MLJ+22], multitask learning [ZY21]

---

[*]equal contribution

38th Conference on Neural Information Processing Systems (NeurIPS 2024).

etc. While numerous methods are available in the literature for training predictive models under distribution shift, uncertainty quantification under distribution shift has received relatively scant attention despite its crucial importance. One notable exception is conformal prediction under distribution shift; [TFBCR19] proposed a variant of standard conformal inference methods to accommodate test data from a distinct distribution from the training data under the covariate shift. Recently, [GC21] introduced an adaptive conformal inference approach suitable for continuously changing distributions over time. Additionally, quantile regression under distribution shift offers another avenue for addressing uncertainty quantification under distribution shift [ERS+22].

Although few methods exist for constructing prediction intervals under distribution shift, most focus primarily on ensuring coverage guarantee rather than minimizing interval width. This prompts the immediate question:

*Can we generate prediction intervals in the target domain that provide both i) coverage guarantee and ii) minimal width?*

This paper seeks to address this question by leveraging model aggregation techniques [NW15, MNW16, CEN14, Vov15, HKNC14]. Suppose we have $K$ different methods for constructing prediction intervals in the *source* domain. Our proposed approach efficiently combines these methods to produce prediction intervals in the *target* domain with adequate coverage and minimal width. When individual methods are the elementary basis functions, such as the kernel basis, the resulting aggregation is indeed a construction of the prediction interval based on the basis functions. Our methodology draws inspiration primarily from recent work [FGM23] on prediction interval aggregation under the i.i.d. setting. However, a key distinction lies in our focus on *unsupervised domain adaptation*, where we can access labeled samples from the source and unlabeled samples from the target domain. Certain assumptions regarding the similarities between these domains are necessary to facilitate knowledge transfer from the source to the target domain. We explore two types of similarities in this paper: i) *covariate shift*, where we assume that the distribution of the response variable $Y$ given $X$ is consistent across both domains, albeit the distribution of $X$ may differ, and ii) *domain shift*, where we assume that the conditional distribution of $Y$ given $X$ remains unchanged up to a measure-preserving transformation. Covariate shift is a well-explored concept in transfer learning and has also garnered attention in uncertainty quantification. It allows different distributions of $X$ while maintaining identical conditional distributions $Y|X$ across domains. For constructing conformal prediction intervals within this framework, see [TFBCR19, HL23, YKT22, LC21] and references therein. On the other hand, *distribution shift* is more general, allowing both the distribution of $X$ and the conditional distribution of $Y|X$ to differ across domains. Our methods in this context draw upon domain matching principles via transport map, as proposed in [CFT14] and further elaborated in subsequent works like [CFTR16, CFHR17, RHS17], among others. The key assumption is the existence of a measure-preserving/domain-aligning map $T$ from the target to the source domain, such that the conditional distribution of $Y|X$ on the target domain matches $Y|T(X)$ on the source domain, i.e., conditional distributions matches upon domain alignment. The case where the domain-aligning map is the optimal transport map has received considerable attention in the literature, e.g., see [CFT14, CFTR16, CFHR17, XLW+20]. Empirical evidence supports the efficacy of domain alignment through optimal transport maps across various datasets. For instance, in [XLW+20], a variant of this method is applied for domain adaptation in image recognition tasks, such as recognizing similarities between USPS [Hul94], MNIST [LBBH98], and SVHN digit images [NWC+11], as well as between different types of images in the Office-home dataset [VECP17], including artistic and product images. Additionally, in [CFT14], the authors explore the impact of domain alignment via optimal transport maps on the face recognition problem, where different poses give rise to distinct domains. However, most of these works concentrate on training predictors that perform well on the target domain without any guarantee regarding uncertainty quantification. To our knowledge, this is the first work to propose a method with rigorous theoretical guarantees for constructing prediction intervals on the target domain under the domain-aligning assumption within an unsupervised domain adaptation framework. We now summarize our contributions.

**Our Contributions:** This paper introduces a novel methodology for aggregating various prediction methods available on the source domain to construct a unified prediction interval on the target domain under both covariate shift and domain shift assumptions. Our approach is simple and easy to implement and requires solving a convex optimization problem, which can even be

simplified to a linear program problem in certain scenarios. We also establish rigorous theoretical guarantees, presenting finite sample concentration bounds to demonstrate that our method achieves adequate coverage with a small width. Furthermore, our methodology extends beyond model aggregation; it can be used to construct efficient prediction intervals from any convex collection of candidate functions. In the paper, we adopt this broader perspective, discussing how the aggregation of prediction intervals emerges as a particular case. Lastly, we validate the effectiveness of our approach by analyzing real-world datasets.

We also want to highlight the differences between our method and a related method proposed in [FGM23]. We deal with unsupervised domain adaptation, i.e., we do not observe any label from the target domain, in contrast to [FGM23], which only deals with i.i.d. data. Hence, significant changes in methodology are required to address the domain shift. Furthermore, as pointed out in Section 3, the shift may cause the optimization problem non-convex, for which we need to introduce a convex surrogate (e.g., the hinge function), leading to additional theoretical challenges.

## 2  Notations and preliminaries

**Notation**   The covariates of the source and the target domains are denoted by $\mathcal{X}_S$ and $\mathcal{X}_T$, respectively, and $\mathcal{X} := \mathcal{X}_S \cup \mathcal{X}_T$. The space of the label is denoted by $\mathcal{Y}$. We use the notation $\mathbb{E}_S$ (resp. $\mathbb{E}_T$) to denote the expectation with respect to the source (resp. target) distribution. The expectation with respect to sample distribution is denoted by $\mathbb{E}_{n,S}$ and $\mathbb{E}_{n,T}$. We use $p_S$ (resp. $p_T$) to denote the probability density function of $X$ on the source and the target domain, respectively. Throughout the paper, we use $c$ to denote universal constants, which may vary from line to line.

### 2.1  Problem formulation

Our setup aligns with the unsupervised domain adaption; we assume to have $n_S$ i.i.d. labeled samples $\{X_{S,i}, Y_{S,i}\}_{i=1}^{n_S} \sim \mathbb{P}_S(X, Y)$ from the source domain, and $n_T$ i.i.d. unlabeled samples $\{X_{T,i}\}_{i=1}^{n_T} \sim \mathbb{P}_T(X)$ from the target domain. Given any $\alpha > 0$, ideally, we want to construct a valid prediction interval with minimal width on the target domain:

$$\min_{u,l} \quad \mathbb{E}_T[u(X) - l(X)], \ \ \text{s.t.} \ \ \mathbb{P}_T\left(l(X) \le Y \le u(X)\right) \ge 1 - \alpha \,. \tag{2.1}$$

In many practical contexts, the preferred prediction interval takes the form of $m(X) \pm g(X)$, where $m(X)$ is a predictor for $Y$ given $X$ (an estimator of $\mathbb{E}_T[Y \mid X]$), and $g(X)$ gauges the uncertainty of the predictor $m(X)$. The optimizer of (2.1) takes this simplified form when the distribution of $Y - \mathbb{E}_T[Y \mid X]$ is symmetric around 0. Moreover, it offers a straightforward interpretation as the pair $(m, g)$ is a predictor and a function quantifying its uncertainty. Within the framework of this simplified prediction interval, we need to estimate $m$ and $g$. Estimating the conditional mean function $m$ is relatively easy and has been extensively studied; one may use any suitable parametric/non-parametric method. Upon estimating $m$, we need to estimate $g$ so that the prediction interval $[m(X) \pm g(X)]$ has both adequate coverage and minimal width. This translates into solving the following optimization problem:

$$\min_{f \in \mathcal{F}} \quad \mathbb{E}_T[f(X)], \ \ \text{s.t.} \ \ \mathbb{P}_T\left((Y - m(X))^2 > f(X)\right) \le \alpha \,. \tag{2.2}$$

Let $f_0$ be the solution of the above optimization problem. Then the optimal prediction interval is $[m_0(x) \pm \sqrt{f_0(x)}]$. However, the key challenge here is that we do not observe the response variable $Y$ from the target, and consequently, solving (2.2) becomes infeasible. Hence, we must rely on transferring our knowledge acquired from labeled observations in the source domain, which necessitates making certain assumptions regarding the similarity between the two domains. Depending on the nature of these assumptions regarding domain similarity, our findings are presented in two sections: Section 3 addresses covariate shift under the bounded density ratio assumption, while Section 4 considers a more general distribution assumption under measure-preserving transformations. Furthermore, as will be shown later, this problem, though well-defined, is not easily implementable. Therefore, we propose a surrogate convex optimization problem in this paper and provide its theoretical guarantees.

### 2.2  Complexity measure

The complexity of the function class $\mathcal{F}$ is usually quantified through the Rademacher complexity, defined as follows.

**Definition 2.1** (Rademacher complexity)**.** *Let $\mathcal{F}$ be a function class and $\{X_i\}_{i=1}^n$ be a set of samples drawn i.i.d. from a distribution $\mathcal{D}$. The Rademacher complexity of $\mathcal{F}$ is defined as*

$$\mathcal{R}_n(\mathcal{F}) = \mathbb{E}_{\epsilon,\mathcal{D}}\left[\sup_{f\in\mathcal{F}}\frac{1}{n}\sum_{i=1}^n \epsilon_i f(X_i)\right], \tag{2.3}$$

*where $\{\epsilon_i\}_{i=1}^n$ are i.i.d. Rademacher random variables that equals to $\pm 1$ with probability $1/2$ each.*

## 3 Covariate shift with bounded density ratio

**Setup and methodology** In this section, we focus on the covariate shift problems, where the marginal densities $p_S(X)$ and $p_T(X)$ of the covariates may vary between the source and target domains, albeit the conditional distribution $Y|X$ remains the same. Denote by $m_0(x) = \mathbb{E}_T[Y|X = x] = \mathbb{E}_S[Y|X = x]$, the conditional mean function. For the ease of the presentation, we assume $m_0$ is known. If unknown, one may use the labeled source data to estimate it using a suitable parametric/non-parametric estimate (e.g., splines, local polynomial, or deep neural networks), subsequently substituting $m_0$ with $\hat{m}$ in our approach. The density ratio of the source and the target distribution of $X$ is denoted by $w_0(x) := p_T(x)/p_S(x)$. We henceforth assume that the density ratio is uniformly bounded:

**Assumption 3.1.** *There exists $W$ such that $\sup_{x\in\mathcal{X}_S} w_0(x) \leq W$.*

If $w_0$ is known, (2.2) has the following sample level counterpart:

$$\min_{f\in\mathcal{F}} \ \mathbb{E}_{n,T}[f(X)], \ \text{ s.t. } \mathbb{E}_{n,S}\left[w_0(X)\mathbb{1}_{(Y-m_0(X))^2 > f(X)}\right] \leq \alpha, \tag{3.1}$$

which is NP-hard owing to the presence of the indicator function. However, in many practical scenarios, it is observed that the shape of the prediction band does not change much if we change the level of coverage (i.e., $\alpha$); only the bands shrink/expand. Indeed, the true shape determines the average width; if the shape is wrong, then the width of the prediction band is quite likely to be unnecessarily large. Therefore, to obtain a prediction interval with adequate coverage and minimal width, one should first identify the shape of the prediction band and then shrink/expand it appropriately to get the desired coverage. This motivates the following two steps procedure:

**Step 1:** (Shape estimation) Obtain an initial estimate $\hat{f}_{\text{init}}$ by solving (3.1) for $\alpha = 0$ (to capture the shape):

$$\min_{f\in\mathcal{F}} \ \mathbb{E}_{n,T}[f(X)], \ \text{ s.t. } f(X_i) \geq (Y_i - m_0(X_i))^2 \ \forall\, 1 \leq i \leq n_S : w_0(X_i) > 0. \tag{3.2}$$

**Step 2:** (Shrinkage) Refine $\hat{f}_{\text{init}}$ by scaling it down using $\hat{\lambda}(\alpha)$, defined as:

$$\hat{\lambda}(\alpha) = \inf\left\{\lambda \geq 0 : \mathbb{E}_{n,S}[w_0(X)\mathbb{1}_{(Y-m_0(X))^2 > \lambda\hat{f}_{\text{init}}(X)}] \leq \alpha\right\}. \tag{3.3}$$

The final prediction interval is:

$$\widehat{\text{PI}}_{1-\alpha}(x) = \left[m_0(x) - \sqrt{\hat{\lambda}(\alpha)\hat{f}_{\text{init}}(x)}, m_0(x) + \sqrt{\hat{\lambda}(\alpha)\hat{f}_{\text{init}}(x)}\right]. \tag{3.4}$$

In Step 1, we relax (3.1) by effectively setting $\alpha = 0$. This relaxation aids in determining the optimal shape while also converting (3.1) into a convex optimization problem (equation (3.2)) as long as $\mathcal{F}$ is a convex collection of functions. Furthermore, in (3.2), we only consider those source observations for which $w_0(x) > 0$, as otherwise, the samples are not informative for the target domain. In practice, $w_0$ is typically unknown; one may use the source and target domain covariates to estimate $w_0$. Various techniques are available for estimating the density ratio (e.g., [USS$^+$16, CMSE22, Qin98, GSH$^+$08] and references therein). However, any such estimator $\hat{w}(x)$ can be non-zero for $x$ where $w_0(x) = 0$ due to estimation error. Consequently, $\hat{w}$ may not be efficient in selecting informative source samples. To mitigate this issue, we propose below a modification of (3.2), utilizing a hinge function $h_\delta(t) := \max\{0, (t/\delta) + 1\}$:

$$\min_{f\in\mathcal{F}} \ \mathbb{E}_{n,T}[f(X)]$$
$$\text{subject to } \mathbb{E}_{n,S}[\hat{w}(X)h_\delta\left((Y-m_0(X))^2 - f(X)\right)] \leq \epsilon, \tag{3.5}$$

with $\delta$ and $\epsilon$ should be chosen based on sample size $n_S$ and the estimation accuracy of $\hat{w}$. When $\hat{w} = w_0$ (i.e., the density ratio is known), then by choosing $\epsilon = 0$ and $\delta \to 0$, (3.5) recovers (3.2). As $h_\delta$ is convex, the optimization problem (3.5) is still a convex optimization problem. We summarize our algorithm in Algorithm 1.

---

**Algorithm 1** Prediction intervals with bounded density ratio

---

1: **Input:** $m_0$ (or $\hat{m}$ if unknown), density ratio estimator $\hat{w}$, function class $\mathcal{F}$, sample $\mathcal{D}_S = \{(X_{S,i}, Y_{S,i})\}_{i=1}^{n_S}$ and $\mathcal{D}_T = \{X_{T,i}\}_{i=1}^{n_T}$, parameters $\delta, \epsilon$, coverage level $1 - \alpha$.

2: Obtain $\hat{f}_{\text{init}}$ by solving (3.5).

3: Obtain the shrink level $\hat{\lambda}(\alpha)$ by solving (3.3) with $w_0$ replaced by $\hat{w}$.

4: **Output:** $\widehat{\text{PI}}_{1-\alpha}(x)$ defined in (3.4).

---

**Theoretical results** We next present theoretical guarantees of the prediction interval obtained via Algorithm 1. For technical convenience, we resort to data-splitting; we divide the source data into two equal parts ($\mathcal{D}_{S,1}$ and $\mathcal{D}_{S,2}$), use $\mathcal{D}_{S,1}$ and $\mathcal{D}_T$ to solve (3.5), and $\mathcal{D}_{S,2}$ to obtain the shrink level $\hat{\lambda}(\alpha)$. Without loss of generality, we assume $m_0 \equiv 0$ (otherwise, we set $Y \leftarrow Y - m_0(X)$). A careful inspection of Step 1 reveals that $\hat{f}_{\text{init}}$ aims to approximate a function $f^*$ defined as follows:

$$f^* = \arg\min_{f \in \mathcal{F}} \mathbb{E}_T[f(X)] \text{ subject to } Y^2 < f(X) \text{ almost surely on target domain}. \quad (3.6)$$

In other words, $\hat{f}_{\text{init}}$ estimates $f^*$ that has minimal width among all functions covering the response variable. This is motivated by the philosophy that the *right shape leads to a smaller width*. The following theorem provides a finite sample concentration bound on the approximation error of $\hat{f}_{\text{init}}$:

**Theorem 3.2.** *Suppose $Y^2 - f^*(X) \leq B$ on the source domain and has a density bounded by $L$. Also assume $\|f\|_\infty \leq B_{\mathcal{F}}$ for all $f \in \mathcal{F}$. Then for*

$$\epsilon \geq L\delta + W\sqrt{\frac{t}{n_S}} + \frac{B + \delta}{\delta} \cdot \left( \mathbb{E}_S\left[|\hat{w}(X) - w_0(X)|\right] + (W + W')\sqrt{\frac{t}{n_S}} \right), \quad (3.7)$$

*we have with probability at least $1 - 3e^{-t}$:*

$$\mathbb{E}_T[\hat{f}_{\text{init}}(X)] \leq \mathbb{E}_T[f^*(X)] + 2\mathcal{R}_{n_T}(\mathcal{F} - f^*) + 2B_{\mathcal{F}}\sqrt{\frac{t}{2n_T}}$$

*where $W' = \|\hat{w}\|_\infty$.*

The bound in the above theorem depends on the Rademacher complexity of $\mathcal{F}$ (the smaller, the better), the estimation error of $w_0$, and an interplay between the choice of $(\epsilon, \delta)$. The lower bound on $\epsilon$ in (3.7) depends on both $\delta$ and $1/\delta$. Although it is not immediate from the above theorem why we need to choose $\epsilon$ to be as small as possible, it will be apparent in our subsequent analysis; indeed if $\epsilon$ is large in (3.5), then $\hat{f}_{\text{init}} \equiv 0$ will be a solution of (3.5). Consequently, the shape will not be captured. Therefore, one should first choose $\delta$ (say $\delta^*$), that minimizes the lower bound (3.7), and then set $\epsilon = \epsilon^*$ equal to the value of the right-hand side of (3.7) with $\delta = \delta^*$, which ensures that $\epsilon^*$ is optimally defined to capture the shape accurately. Once the shape is identified, we shrink it properly in Step 2 to attain the desired coverage and reduce the width. Although ideally $\hat{\lambda}(\alpha) \leq 1$, it is not immediately guaranteed as we use separate data ($\mathcal{D}_{S,2}$) for shrinking. The following lemma shows that $\hat{\lambda}(\alpha) \leq 1$ for any fixed $\alpha > 0$ as long as the sample size is large enough. Recall that the data were split into exactly half with size $n_S = |\mathcal{D}_S|$.

**Lemma 3.3.** *Under the aforementioned choice of $(\epsilon^*, \delta^*)$, we have with high probability:*

$$\frac{1}{n_S/2} \sum_{i \in \mathcal{D}_{S,2}} \hat{w}(X_i) \mathbb{1}_{\{(Y_i - m_0(X_i))^2 > \hat{f}_{\text{init}}(X_i)\}} \leq \alpha,$$

*for all large $n_S$, provided that $\hat{w}$ is a consistent estimator of $w_0$. Hence, $\hat{\lambda}(\alpha) \leq 1$.*

Our final theorem for this section provides a coverage guarantee for the prediction interval given by Algorithm 1.

**Theorem 3.4.** *For the prediction interval obtained in (3.4), with probability greater than $1 - 2e^{-t}$:*

$$\left| \mathbb{P}_T\left( Y^2 > \hat{\lambda}(\alpha)\hat{f}_{\text{init}}(X) \mid \mathcal{D}_S \cup \mathcal{D}_T \right) - \alpha \right| \leq \mathbb{E}_S\left[|\hat{w}(X) - w(X)|\right] + (2W + W')\sqrt{\frac{t}{2n_S}} + \sqrt{\frac{C}{n_S}}$$

*for some constant $C > 0$ and $W' = \|\hat{w}\|_\infty$.*

Theorem 3.4 validates the coverage of the prediction interval derived through Algorithm 1, achieving the desired coverage level as the estimate of $w_0$ improves and sample size expands. Theorems 3.2 and 3.4 collectively demonstrate the efficacy of our method in maintaining validity and accurately capturing the optimal shape of the prediction band, which in turn leads to small interval widths.

**Remark 3.5.** *In our optimization problem, we've substituted the indicator loss with the hinge loss function to ensure convexity. However, it's worth noting that if we know the subset of $\mathcal{X}_S$ where $w_0(x) > 0$ beforehand, we could directly optimize (3.2). This approach would be easy to implement and wouldn't involve tuning parameters $(\delta, \epsilon)$. A special case is when $w_0(x) > 0$ for all $x \in \mathcal{X}_S$ (as is true in our experiment), which simplifies the condition in (3.2) to $f(X_i) \geq (Y_i - m_0(X_i))^2$ for all $1 \leq i \leq n_S$. However, if this information is unavailable, one can still employ (3.2) by enforcing the constraint on all source observations. While this approach might result in wider prediction intervals, it is easy to implement and doesn't require tuning parameters.*

## 4 Domain shift and transport map

**Setup and methodology**   In the previous section, we assume a uniform bound on the density ratio. However, this may not be the case in reality; it is possible that there exists $x \in \mathsf{supp}(\mathcal{X}_T) \cap \mathsf{supp}(\mathcal{X}_S^c)$, which immediately implies that $w_0(x) = \infty$. In image recognition problems, if the source data are images taken during the day at some place, and the target data are images taken at night, then this directly results in an unbounded density ratio (due to the change in the background color). Yet a transport map could effectively model this shift by adapting features from the source to correspond with those of the target, maintaining the underlying patterns or object recognition capabilities across both domains. To perform transfer learning in this setup, we model the domain shift via a measure transport map $T_0$ that preserves the conditional distribution, as elaborated in the following assumption:

**Assumption 4.1.** *There exists a measure transport map $T_0 : \mathcal{X}_T \rightarrow \mathcal{X}_S$, i.e., $T_0(X_T) \stackrel{d}{=} X_S$, such that: $\mathbb{P}_T(Y \mid X = x) \stackrel{d}{=} \mathbb{P}_S(Y \mid X = T_0(x)), \ \forall x \in \mathcal{X}_T$.*

This assumption allows the extrapolation of source domain information to the target domain via $T_0$, enabling the construction of prediction intervals at $x \in \mathcal{X}_T$ by leveraging the analogous intervals at $T_0(x) \in \mathcal{X}_S$. Inspired by this observation, we present our methodology in Algorithm 2 that essentially consists of two key steps: i) constructing a prediction interval in the source domain and ii) transporting this interval to the target domain using the estimated transport map $T_0$. If $T_0$ (or its estimate) is not given, it must be estimated from the source and the target covariates. Various methods are available in the literature (e.g., [DNWP22, SDF$^+$17, MTOL20, DGS21]), and practitioners can pick a method at their convenience. Notably, the processes described in equations (4.1) and (4.2) follow the methodology (i.e., (3.2) and (3.3)) from Section 3 for scenarios without shift (i.e., $w_0 \equiv 1$), adding a slight $\delta$ to ensure coverage even when $\mathcal{F}$ is complex. In Algorithm 2, we assume

---

**Algorithm 2** Transport map

1: **Input:** conditional mean function $m_0$ on the source domain, transport map estimator $\hat{T}_0$, function class $\mathcal{F}$, sample $\mathcal{D}_S = \{(X_{S,i}, Y_{S,i})\}_{i=1}^{n_S}$ and $\mathcal{D}_T = \{X_{T,i}\}_{i=1}^{n_T}$, parameter $\delta$, coverage level $1 - \alpha$.

2: Obtain $\hat{f}_{\text{init}}$ by solving:
$$\min_{f \in \mathcal{F}} \ \frac{1}{n_S} \sum_{i=1}^{n_S} f(X_{S,i}), \ \text{s.t.} \ f(X_{S,i}) \geq (Y_{S,i} - m_0(X_{S,i}))^2 \ \forall \, i \in [n_S]. \tag{4.1}$$

3: Obtain the shrink level
$$\hat{\lambda}(\alpha) := \inf \left\{ \lambda > 0 : \frac{1}{n_S} \sum_{i=1}^{n_S} \mathbb{1}_{(Y_{S,i} - m_0(X_{S,i}))^2 \geq \lambda(\hat{f}_{\text{init}}(X_{S,i}) + \delta)} \leq \alpha \right\}. \tag{4.2}$$

4: **Output:** $\widehat{\text{PI}}_{1-\alpha}(x) = \left[ m_0 \circ \hat{T}_0(x) \pm \sqrt{\hat{\lambda}(\alpha) \cdot \left( \hat{f}_{\text{init}} \circ \hat{T}_0(x) + \delta \right)} \right].$

---

the conditional mean function $m_0$ on the source domain is known. In cases where the conditional

mean function $m_0$ on the source domain is unknown, it can be estimated using standard regression methods from labeled source data, after which $m_0$ is replaced by this estimate, $\hat{m}$.

**Remark 4.2** (Model aggregation). *Suppose we have $K$ different methods $\{f_1, \ldots, f_K\}$ for constructing prediction intervals in the source domain. In the context of model aggregation, (4.1) then reduces to:*

$$\min_{\alpha_1, \ldots, \alpha_K} \quad \frac{1}{n_S} \sum_{i=1}^{n_S} \Big\{ \sum_{j=1}^{K} \alpha_j f_j(X_{S,i}) \Big\}$$

$$\text{subject to} \quad \sum_{j=1}^{K} \alpha_j f_j(X_{S,i}) \geq (Y_{S,i} - m_0(X_{S,i}))^2 \ \forall \, i \in [n_S] \,,$$

$$\alpha_j \geq 0, \quad \forall \, 1 \leq j \leq K \,.$$

*In other words, the function class $\mathcal{F}$ is a linear combination of the candidate methods. The problem is then simplified to a linear program problem, which can be implemented efficiently using standard solvers.*

**Theoretical results**   We now present theoretical guarantees of our methodology to ensure that our method delivers what it promises: a prediction interval with adequate coverage and small width. For technical simplicity, we split data here: divide the labeled source observation with two equal parts (with $n_S/2$ observations in each), namely $\mathcal{D}_{S,1}$ and $\mathcal{D}_{S,2}$. We use $\mathcal{D}_{S,1}$ to solve (4.1) and obtain the initial estimator $\hat{f}_{\text{init}}$, and $\mathcal{D}_{S,2}$ to solve (4.2), i.e. obtaining the shrinkage factor $\hat{\lambda}(\alpha)$. Henceforth, without loss of generality, we assume $m_0 = 0$ and present the theoretical guarantees of our estimator. We start with an analog of Theorem 3.2, which ensures that with high probability $\hat{f}_{\text{init}} \circ \hat{T}_0$ approximates the function that has minimal width among all the functions in $\mathcal{F}$ composed with $T_0$ that covers the labels on the target almost surely:

**Theorem 4.3.** *Assume the function class $\mathcal{F}$ is $B_{\mathcal{F}}$-bounded and $L_{\mathcal{F}}$-Lipschitz. Define*

$$\Delta = \min \big\{ \mathbb{E}_T[f \circ T_0(X)] : f \in \mathcal{F}, Y^2 \leq f \circ T_0(X) \text{ a.s. on target domain} \big\} \,.$$

*Then we have with probability $\geq 1 - e^{-t}$:*

$$\mathbb{E}_T[\hat{f}_{\text{init}} \circ \hat{T}_0(X)] \leq \Delta + 4\mathcal{R}_{n_S}(\mathcal{F}) + L_{\mathcal{F}} \mathbb{E}_T[|\hat{T}_0(X) - T_0(X)|] + 4B_{\mathcal{F}} \sqrt{\frac{t}{2n_S}} \,.$$

The upper bound on the population width of $\hat{f}_{\text{init}} \circ \hat{T}_0(x)$ consists of four terms: the first term is the *minimal possible width* that can be achieved using the functions from $\mathcal{F}$, the second term involves the Rademacher complexity of $\mathcal{F}$, the third term encodes the estimation error of $T_0$, and the last term is the deviation term that influences the probability. Hence, the margin between the width of the predicted interval and the minimum achievable width is small, with the convergence rate relying on the precision of estimating $T_0$ and the complexity of $\mathcal{F}$, as expected.

We next establish the coverage guarantee of our estimator of Algorithm 2, obtained upon suitable truncation of $\hat{f}_{\text{init}}$. As mentioned, the shrinkage operation is performed on a separate dataset $\mathcal{D}_{S,2}$. Therefore, it is not immediate whether the shrinkage factor $\hat{\lambda}(\alpha)$ is smaller than 1, i.e., whether we are indeed shrinking the confidence interval ($\hat{\lambda}(\alpha) > 1$ is undesirable, as it will widen $\hat{f}_{\text{init}}$, increasing the width of the prediction band). The following lemma shows that with high probability, $\hat{\lambda}(\alpha) \leq 1$.

**Lemma 4.4.** *With probability greater than or equal to $1 - e^{-t}$, we have:*

$$\mathbb{P}(\hat{\lambda}(\alpha) > 1 \mid \mathcal{D}_{S,1}, \mathcal{D}_T) \leq e^{-\frac{(\alpha - p_{n_S})^2 n_S}{6 p_{n_S}}} \,,$$

*where*

$$p_{n_S} = \mathbb{P}_S \left( Y^2 \geq \hat{f}_{\text{init}}(X) + \delta \,\big|\, \mathcal{D}_{S,1}, \mathcal{D}_T \right) \leq \tfrac{4}{\delta} \left( \sqrt{\frac{\mathbb{E}_S[Y^4]}{n_S}} + \mathcal{R}_{n_S}(\mathcal{F}) \right) + \sqrt{\frac{t}{n_S}} \,.$$

Here $p_{n_S}$ is the conditional probability of a test observation $Y$ falling outside $[-\sqrt{\hat{f}_{\text{init}}(X) + \delta}, \sqrt{\hat{f}_{\text{init}}(X) + \delta}]$, which is small as evident from the above lemma. In particular, for model aggregation, if $\mathcal{F}$ is the linear combination of $K$ functions, then $p_{n_S}$ is of the order $\sqrt{K/n_S}$. Hence, the final prediction interval is guaranteed to be a compressed form of $\hat{f}_{\text{init}}$ with an overwhelmingly high probability. We present our last theorem of this section, confirming that the prediction interval derived from Algorithm 2 achieves the intended coverage level with a high probability:

**Theorem 4.5.** *Under the same setup of Theorem 4.3, along with the assumption that $f_S(y \mid x)$ is uniformly bounded by $G$, we have with probability greater than $1 - cn_S^{-10}$ that*

$$\left| \mathbb{P}_T \left( Y^2 \geq \hat{\lambda}(\alpha) \left( \hat{f}_{\text{init}} \circ \hat{T}_0(X) + \delta \right) \mid \mathcal{D}_S \cup \mathcal{D}_T \right) - \alpha \right|$$
$$\leq C\sqrt{\frac{\log n_S}{n_S}} + GL_{\mathcal{F}} \cdot \mathbb{E}_T \left[ \left| \hat{T}_0(X) - T_0(X) \right| \right].$$

As for Theorem 4.3, the bound obtained in Theorem 4.5 also depends on two crucial terms: Rademacher complexity of $\mathcal{F}$ and estimation error of $T_0$. Therefore, the key takeaway of our theoretical analysis is that the prediction interval obtained from Algorithm 2 asymptotically achieves nominal coverage guarantee and minimal width. Furthermore, the approximation error intrinsically depends on the Rademacher complexity of the underlying function class and the precision in estimating $T_0$.

**Remark 4.6** (Measure preserving transformation)**.** *In our approach, $T_0$ is employed to maintain measure transformation, although it may not necessarily be an optimal transport map. Yet, estimating $T_0$ can be challenging in many practical scenarios. In such cases, simpler transformations like linear or quadratic adjustments are often utilized to align the first few moments of the distributions. Various methods provide such simple solutions, including, but not limited to, CORAL [SFS17] and ADDA [THSD17].*

## 5 Application

In this section, we illustrate the effectiveness of our method by applying it to five different datasets: i) airfoil dataset [DG19], ii) real estate data [Yeh18], iii) energy efficiency data [TX12b], iv) appliance energy prediction data [Can17], and v) ET Dataset (ETT-small) [ZZP+21]. The first four datasets are freely available in the UCI repository, and the last dataset can be found in this GitHub link. Here, we illustrate the procedure using the airfoil dataset, and the details of our experiments using the other datasets can be found in Appendix C. The airfoil dataset includes 1503 observations, featuring a response variable $Y$ (scaled sound pressure level) and a five-dimensional covariate $X$ (log of frequency, angle of attack, chord length, free-stream velocity, log of suction side displacement thickness). We assess and compare the performance of our prediction intervals in terms of coverage and width with those generated by the weighted split conformal prediction method described in [TFBCR19]. We use the same data-generating process described in [TFBCR19] to facilitate a direct comparison. We run experiments 200 times; each time, we randomly partition the data into two parts $\mathcal{D}_{\text{train}}$ and $\mathcal{D}_{\text{test}}$, where $\mathcal{D}_{\text{train}}$ contains $75\%$ of the data, and $\mathcal{D}_{\text{test}}$ contains $25\%$ of the data. Following [TFBCR19], we *shift* the distribution of the covariates of $\mathcal{D}_{\text{test}}$ by weighted sampling with replacement, where the weights are proportional to

$$w(x) = \exp(x^T \beta), \quad \text{where} \quad \beta = (-1, 0, 0, 0, 1).$$

These reweighted observations in $\mathcal{D}_{\text{test}}$, which we call $\mathcal{D}_{\text{shift}}$, act as observations from the target domain. Clearly, by our data generation mechanism $w_0(x) = f_T(x)/f_S(x) = c \exp(x^\top \beta)$, where $c$ is the normalizing constant. The source and target domains share the same support under this configuration. As our methodology is developed for unsupervised domain adaptation, we do not use the label information of $\mathcal{D}_{\text{shift}}$ to develop the target domain's prediction interval.

**Density ratio estimation** We use the probabilistic classification technique to estimate the density based on the source and the target covariates. Let $X_1, \ldots, X_{n_1}$ be the covariates in dataset $\mathcal{D}_{\text{train}}$ and $X_{n_1+1}, \ldots, X_{n_1+n_2}$ be the covariates in dataset $D_{\text{shift}}$. The density ratio estimation proceeds in two steps: (1) logistic regression is applied to the feature-class pairs $\{(X_i, C_i)\}_{i=1}^n$, where $C_i = 0$ for

$i = 1, \ldots, n_1$ and $C_i = 1$ for $i = n_1 + 1, \ldots, n_1 + n_2$, yielding an estimate of $\mathbb{P}(C = 1 \mid X = x)$, denoted as $\hat{p}(x)$; (2) the density ratio estimator is then defined as $\hat{w}(x) = \frac{n_1}{n_2} \cdot \frac{\hat{p}(x)}{1 - \hat{p}(x)}$. Further explanations are provided in Appendix B.

**Implementation of our method and results**  As the mean function $m_0(x) = \mathbb{E}[Y \mid X = x]$ (which is the same on the source and the target domain) is unknown, we first estimate it via linear regression, which henceforth will be denoted by $\hat{m}(x)$. To construct a prediction interval, we consider the model aggregation approach, i.e., the function class $\mathcal{F}$ is defined as the linear combination of the following six estimates:

(1) **Estimator 1**$(f_1)$: A neural network based estimator with depth=1, width=10 that estimates the 0.85 quantile function of $(Y - \hat{m}(X))^2 \mid X = x$.

(2) **Estimator 2**$(f_2)$: A fully connected feed forward neural network with depth=2 and width=50 that estimates the 0.95 quantile function of $(Y - \hat{m}(X))^2 \mid X = x$.

(3) **Estimator 3**$(f_3)$: A quantile regression forest estimating the 0.9 quantile function of $(Y - \hat{m}(X))^2 \mid X = x$.

(4) **Estimator 4**$(f_4)$: A gradient boosting model estimating the 0.9 quantile function of $(Y - \hat{m}(X))^2 \mid X = x$.

(5) **Estimator 5**$(f_5)$: An estimate of $\mathbb{E}[(Y - \hat{m}(X))^2 \mid X = x]$ using random forest.

(6) **Estimator 6**$(f_6)$: The constant function 1.

Here, the quantile estimators are obtained by minimizing the corresponding check loss. The implementation of our method is summarized as follows: (1) We divide the training data $\mathcal{D}_{\text{train}}$ into two halves $\mathcal{D}_1 \cup \mathcal{D}_2$. We utilize dataset $\mathcal{D}_1$ to derive a mean estimator and six aforementioned estimates. We also employ the covariates from $\mathcal{D}_1$ and $D_{\text{shift}}$ to compute a density ratio estimator. (2) We further split $\mathcal{D}_2$ into two equal parts $\mathcal{D}_{2,1}$ and $\mathcal{D}_{2,2}$. $\mathcal{D}_{2,1}$, along with covariates from $\mathcal{D}_{\text{shift}}$, is used to find the optimal aggregation of the six estimates to capture the shape, i.e., for obtaining $\hat{f}_{\text{init}}$. The second part $\mathcal{D}_{2,2}$ is used to shrink the interval to achieve $1 - \alpha = 0.95$ coverage, i.e. to estimate $\hat{\lambda}(\alpha)$. (3) We evaluate the effectiveness of our approach in terms of the coverage and average bandwidth on the $D_{\text{shift}}$ dataset.

In Figure 1, we present the histograms of the coverage and the average bandwidth of our method, and a more general version of weighted conformal prediction in [TFBCR19] over 200 experiments (see Appendix B for details), which show that our method consistently yields a shorter prediction interval than the weighted conformal prediction while maintaining coverage. Over 200 experiments, the average coverage achieved by our method was 0.964029 (SD = 0.04), while the weighted conformal prediction method achieved an average coverage of 0.9535 (SD = 0.036). Additionally, the average width of the prediction intervals for our method was 13.654 (SD = 2.22), compared to 20.53 (SD = 4.13) for the weighted conformal prediction. Regarding the performance of intervals over 95% coverage, our method achieved this in 72.5% of cases with an average width of 14.35 (SD = 2.22). In contrast, the weighted conformal prediction method did so in 57% of cases with an average width of 21.4 (SD = 4.39). Boxplots are presented in Appendix B for further comparison.

## 5.1  Robustness of our method

One of the strengths of our approach lies in its resilience to the misspecification of certain components. The core idea behind our method is to combine multiple predictors to create a prediction interval that ensures sufficient coverage while maintaining a narrow average width. These individual prediction intervals may be based on estimators of conditional quantiles, means, variances, and other metrics. If some of the component prediction intervals exhibit poor performance, whether due to inadequate coverage or excessive width, our method typically assigns them lower weights, making it robust to their deficiencies. In contrast, other conformal methods that heavily depend on a single component tend to underperform, particularly with respect to average width. To illustrate this phenomenon, we evaluate our method under model misspecification using a simple one-dimensional simulation setup.

$$X \sim \text{unif}([-1, 1]), \quad \xi \sim \text{unif}([-1, 1]), \quad X \perp\!\!\!\perp \xi$$
$$Y = \sqrt{1 + 25X^4}\,\xi\,.$$

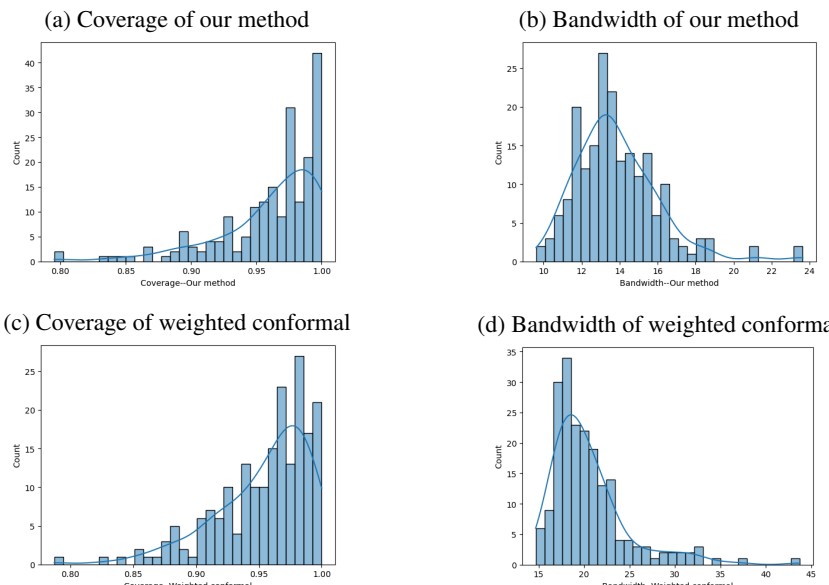

Figure 1: Experiments on Airfoil data using Algorithm 1

| Max depth | Avg. width–Our Method | Avg. width–WVAC |
|---|---|---|
| 3 | 2.07(0.975) | 3.08(0.9712) |
| 5 | 2.07(0.95) | 3.28(0.9664) |
| 7 | 2.068(0.94) | 3.33(0.97) |
| 15 | 2.08(0.97) | 5.00(0.97) |

Table 1: Robustness of our method and WVAC. The number inside the parenthesis is the median of coverage over these Monte Carlo iterations.

As mentioned in the previous subsection, our method aggregates six predictor intervals, including an estimator for the conditional variance function (Estimator 5). The weighted variance-adjusted conformal prediction interval (WVAC) relies on accurately estimating this conditional variance. We estimate the conditional variance using a random forest with varying depths ($\{3, 5, 7, 15\}$). For simulation purpose, we generate $n = 2500$ samples, keeping $75\%$ as source data and resampling the remaining $25\%$ with weighted samples proportional to $w(x) \propto (1 + \exp(-2x))^{-1}$. As depth increases, overfitting leads to poor out-of-sample variance predictions. Table 1 summarizes our findings over 100 Monte Carlo iterations, showing that WVAC's average width increases with depth, while our method's average width remains stable.

## 6    Conclusion

This paper focuses on unsupervised domain shift problems, where we have labeled samples from the source domain and unlabeled samples from the target domain. We introduce methodologies for constructing prediction intervals on the target domain that are designed to ensure adequate coverage while minimizing width. Our analysis includes scenarios in which the source and target domains are related either through a bounded density ratio or a measure-preserving transformation. Our proposed methodologies are computationally efficient and easy to implement. We further establish rigorous finite sample theoretical guarantees regarding the coverage and width of our prediction intervals. Finally, we demonstrate the practical effectiveness of our methodology through its application to the airfoil dataset.

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

# A  Proofs

## A.1  Proof of Theorem 3.2

First, we show that for our choice of $(\epsilon, \delta)$, as depicted in Theorem 3.2, $f^*$ is a feasible solution of equation (3.5). Consider $w_0$ instead of $\hat{w}$. By definition of $f^*$,

$$\mathbb{P}_T(Y^2 \leq f^*(X)) = 1 \iff \mathbb{E}_S\left[w_0(X)\mathbb{1}_{Y^2 > f^*(X)}\right] = 0$$
$$\iff w_0(X)\mathbb{1}_{Y^2 > f^*(X)} = 0 \quad \text{a.s. on the source domain}.$$

This implies:

$$\frac{1}{n_S/2} \sum_{i \in \mathcal{D}_{S,1}} w_0(X_i)h_\delta\left(Y_i^2 - f^\star(X_i)\right)$$

$$= \frac{1}{n_S/2} \sum_{i \in \mathcal{D}_{S,1}} w_0(X_i)h_\delta\left(Y_i^2 - f^\star(X_i)\right)\mathbb{1}_{Y_i^2 \leq f^\star(X_i)}$$

$$= \frac{1}{n_S/2} \sum_{i \in \mathcal{D}_{S,1}} w_0(X_i)h_\delta\left(Y_i^2 - f^\star(X_i)\right)\mathbb{1}_{f^\star(X_i) - \delta \leq Y_i^2 \leq f^\star(X_i)}$$

$$\leq \frac{1}{n_S/2} \sum_{i \in \mathcal{D}_{S,1}} w_0(X_i)\mathbb{1}_{f^\star(X_i) - \delta \leq Y_i^2 \leq f^\star(X_i)},$$

where the first equality follows from the fact that $w_0(X)\mathbb{1}_{Y^2 > f^\star(X)} = 0$ a.s. on the source domain, the second equality follows from the fact that $h_\delta(t)\mathbb{1}_{t < -\delta} = 0$ for all $t$, and the last inequality follows from the fact that $h_\delta(Y_i^2 - f^\star(X_i)) \leq 1$ when $Y_i^2 - f^\star(X_i) \leq 0$. Since $w_0(X)\mathbb{1}_{f^\star(X) - \delta \leq Y^2 \leq f^\star(X)} \leq W$, by Hoeffding's inequality, we have with probability at least $1 - e^{-t}$:

$$\frac{1}{n_S/2} \sum_{i \in \mathcal{D}_{S,1}} w_0(X_i)h_\delta\left(Y_i^2 - f^\star(X_i)\right) \leq \mathbb{E}_S\left[w_0(X)\mathbb{1}_{f^\star(X) - \delta \leq Y^2 \leq f^\star(X)}\right] + W\sqrt{\frac{t}{n_S}}$$

$$= \mathbb{P}_T\left(f^\star(X) - \delta \leq Y^2 \leq f^\star(X)\right) + W\sqrt{\frac{t}{n_S}}$$

$$\leq L\delta + W\sqrt{\frac{t}{n_S}},$$

where $L$ is upper bound on the density of $Y^2 - f^*(X)$. Call this event $\Omega_1$ that the above bound holds. At this event we have:

$$\frac{1}{n_S/2} \sum_{i \in \mathcal{D}_{S,1}} \hat{w}(X_i)h_\delta\left(Y_i^2 - f^\star(X_i)\right)$$

$$= \frac{1}{n_S/2} \sum_{i \in \mathcal{D}_{S,1}} w_0(X_i)h_\delta\left(Y_i^2 - f^\star(X_i)\right) + \frac{1}{n_S/2} \sum_{i \in \mathcal{D}_{S,1}} (\hat{w}(X_i) - w_0(X_i))h_\delta\left(Y_i^2 - f^\star(X_i)\right)$$

$$\leq L\delta + W\sqrt{\frac{t}{n_S}} + \frac{B + \delta}{\delta} \cdot \frac{2}{n_S} \sum_{i=1}^{n_S/2} |\hat{w}(X_i) - w_0(X_i)|,$$

where the last inequality follows from the fact that $h_\delta(t) \leq (B+\delta)/\delta$ if $t \leq B$. Finally, to bound the last summand, we again apply Hoeffding's inequality. As $\|\hat{w}\|_\infty \leq W'$, we have with probability greater than or equal to $1 - e^{-t}$:

$$\frac{1}{n_S/2} \sum_{i=1}^{n_S/2} |\hat{w}(X_i) - w_0(X_i)| \leq \mathbb{E}_S\left[|\hat{w}(X) - w_0(X)|\right] + (W + W')\sqrt{\frac{t}{n_S}}.$$

If we denote the event $\Omega_2$ where the above inequality holds, then on the event $\Omega_1 \cap \Omega_2$, we have:

$$\frac{1}{n_S/2} \sum_i \hat{w}(X_i)h_\delta\left(Y_i^2 - f^\star(X_i)\right)$$

$$\leq L\delta + W\sqrt{\frac{t}{n_S}} + \frac{B + \delta}{\delta} \cdot \left(\mathbb{E}_S\left[|\hat{w}(X) - w_0(X)|\right] + (W + W')\sqrt{\frac{t}{n_S}}\right) \leq \epsilon.$$

Furthermore,
$$\mathbb{P}(\Omega_1 \cap \Omega_2) \geq \mathbb{P}(\Omega_1) + \mathbb{P}(\Omega_2) - 1 \geq 1 - 2e^{-t}.$$
Therefore, we conclude that with probability $\geq 1 - 2e^{-t}$, $f^*$ is a feasible solution.

We now proof Theorem 2.2 on the event $\Omega_1 \cap \Omega_2$, when $f^*$ is a feasible solution. Then we have, $\mathbb{P}_{n,T}(\hat{f}_{\text{init}}(X)) \leq \mathbb{P}_{n,T}(f^*(X))$ on this event, by the optimality of $\hat{f}_{\text{init}}$ in equation (3.5). Then we have:

$$\begin{aligned}
\mathbb{E}_T[\hat{f}_{\text{init}}(X)] &= \mathbb{P}_{n_T}(\hat{f}_{\text{init}}(X)) + (\mathbb{P}_T - \mathbb{P}_{n_T})(\hat{f}_{\text{init}}(X)) \\
&\leq \mathbb{P}_{n_T}(f^*(X)) + (\mathbb{P}_T - \mathbb{P}_{n_T})(\hat{f}_{\text{init}}(X)) \\
&= \mathbb{E}_T[f^*(X)] + (\mathbb{P}_{n_T} - \mathbb{P}_T)(f^*(X) - \hat{f}_{\text{init}}(X)) \\
&\leq \mathbb{E}_T[f^*(X)] + \sup_{f \in \mathcal{F}} |(\mathbb{P}_{n_T} - \mathbb{P}_T)(f^*(X) - f(X))|
\end{aligned}$$

Finally as $f - f^*$ is upper bounded by $F' = B_{\mathcal{F}} + \|f^*\|_\infty$ (as $f$ is uniformly upper bounded by F). Therefore, by Mcdiarmid's inequality, we have with probability $1 - e^{-t}$:

$$\sup_{f \in \mathcal{F}} |(\mathbb{P}_{n_T} - \mathbb{P}_T)(f^*(X) - f(X))| \leq \mathbb{E}_T \left[ \sup_{f \in \mathcal{F}} |(\mathbb{P}_{n_T} - \mathbb{P}_T)(f^*(X) - f(X))| \right] + F' \sqrt{\frac{t}{2n_T}}.$$

Call this event $\Omega_3$. Furthermore, by standard symmetrization:

$$\mathbb{E}_T \left[ \sup_{f \in \mathcal{F}} |(\mathbb{P}_{n_T} - \mathbb{P}_T)(f^*(X) - f(X))| \right] \leq 2\mathcal{R}_{n_T}(\mathcal{F} - f^*),$$

where $\mathcal{R}_{n_T}(\mathcal{F} - f^*)$ is the Rademacher complexity of $\mathcal{F} - f^*$. Therefore, on $\cap_{i=1}^3 \Omega_i$, we have:

$$\mathbb{E}_T[\hat{f}_{\text{init}}(X)] \leq \mathbb{E}_T[f^*(X)] + 2\mathcal{R}_{n_T}(\mathcal{F} - f^*) + F' \sqrt{\frac{t}{2n_T}},$$

and $\mathbb{P}(\cap_{i=1}^3 \Omega_i) \geq 1 - 3e^{-t}$. This completes the proof.

## A.2 Proof of Lemma 3.3

We prove the lemma into two steps; first we show that $\hat{f}_{\text{init}}$ satisfies $\mathbb{P}_T(Y^2 > \hat{f}_{\text{init}}(X)) \leq \tau$ with high probability for some small $\tau$. Next we argue that, on $\mathcal{D}_{S,2}$, we have $(2/n_S) \cdot \sum_{i \in \mathcal{D}_{S,2}} \hat{w}(X_i)\mathbb{1}(Y_i^2 \geq \hat{f}_{\text{init}}(X_i)) \leq \check{\tau}$ with high probability for some small $\check{\tau}$. Then as long as $\check{\tau} \leq \alpha$, we conclude the proof of the lemma.

**Step 1:** Note that, by feasibility, $\hat{f}_{\text{init}}$ satisfies:

$$\frac{1}{n_S/2} \sum_{i \in \mathcal{D}_{S,1}} \hat{w}(X_i)h_\delta(Y_i^2 - \hat{f}_{\text{init}}(X_i)) \leq \epsilon.$$

This implies:

$$\begin{aligned}
&\mathbb{E}_T \left[ h_\delta \left( Y^2 - \hat{f}_{\text{init}}(X) \right) \right] \\
&= \mathbb{E}_S \left[ w_0(X)h_\delta \left( Y^2 - \hat{f}_{\text{init}}(X) \right) \right] \\
&= \frac{1}{n_S/2} \sum_{i \in \mathcal{D}_{S,1}} w_0(X_i)h_\delta(Y_S^2 - \hat{f}_{\text{init}}(X_i)) + (\mathbb{P}_S - \mathbb{P}_{n_S/2}) w_0(X)h_\delta(Y^2 - \hat{f}_{\text{init}}(X)) \\
&= \frac{1}{n_S/2} \sum_{i \in \mathcal{D}_{S,1}} \hat{w}(X_i)h_\delta(Y_i^2 - \hat{f}_{\text{init}}(X_i)) + \frac{1}{n_S/2} \sum_{i \in \mathcal{D}_{S,1}} (w_0(X_i) - \hat{w}(X_i))h_\delta(Y_i^2 - \hat{f}_{\text{init}}(X_i)) \\
&\qquad + (\mathbb{P}_S - \mathbb{P}_{n_S/2}) w_0(X)h_\delta(Y^2 - \hat{f}_{\text{init}}(X)) \\
\\
&\leq \epsilon + \frac{B+\delta}{\delta} \|\hat{w} - w_0\|_{L_1(\mathbb{P}_{n_1,S})} + \sup_{f \in \mathcal{F}} \left| (\mathbb{P}_S - \mathbb{P}_{n_S/2}) w_0(X)h_\delta(Y^2 - f(X)) \right|
\end{aligned}$$

Now, as $h_\delta(Y^2 - f(X)) \leq (B+\delta)/\delta$ and $w_0 \leq W$, we have by Mcdiarmid's inequality, with probability $\geq 1 - e^{-t}$:

$$\sup_{f \in \mathcal{F}} \left| \left( \mathbb{P}_S - \mathbb{P}_{n_S/2} \right) w_0(X) h_\delta(Y^2 - f(X)) \right|$$

$$\leq \mathbb{E}_S \left[ \sup_{f \in \mathcal{F}} \left| \left( \mathbb{P}_S - \mathbb{P}_{n_S/2} \right) w_0(X) h_\delta(Y^2 - f(X)) \right| \right] + W \frac{B+\delta}{\delta} \sqrt{\frac{t}{n_S}}$$

$$\leq 2 \mathcal{R}_{n_S/2, \mathcal{F}}(w_0 h_\delta \circ f) + W \frac{B+\delta}{\delta} \sqrt{\frac{t}{n_S}} \, .$$

Meanwhile, as in the proof of Theorem 3.2, with probability $\geq 1 - e^{-t}$:

$$\|\hat{w} - w_0\|_{L_1(\mathbb{P}_{n_1, S})} \leq \mathbb{E}_S \left[ |\hat{w}(X) - w(X)| \right] + (W + W') \sqrt{\frac{t}{n_S}} \, .$$

Choosing $t = 10 \log n_S$ we obtain that with probability $\geq 1 - 2 n_S^{-10}$:

$$\mathbb{E}_T \left( h_\delta \left( Y_T^2 - \hat{f}_{\text{init}}(X_T) \right) \right)$$

$$\leq \epsilon + \frac{B+\delta}{\delta} \left( \mathbb{E}_S \left[ |\hat{w}(X) - w_0(X)| \right] + (W + W') \sqrt{\frac{10 \log n_S}{n_S}} \right)$$

$$+ 2 \mathcal{R}_{n_S/2, \mathcal{F}}(w_0 h_\delta \circ f) + W \frac{B+\delta}{\delta} \sqrt{\frac{10 \log n_S}{n_S}}$$

$$\leq \epsilon + \frac{B+\delta}{\delta} \left( \mathbb{E}_S \left[ |\hat{w}(X) - w_0(X)| \right] + (2W + W') \sqrt{\frac{10 \log n_S}{n_S}} \right) + 2 \mathcal{R}_{n_S/2, \mathcal{F}}(w_0 h_\delta \circ f) \, .$$

We next bound the Rademacher complexity of $\mathcal{R}_{n_S/2, \mathcal{F}}(w_0 h_\delta \circ f)$. By symmetrization, we have with $\zeta_1, \ldots \zeta_{n_S/2}$ i.i.d. Rademacher(1/2):

$$\mathcal{R}_{n_S/2, \mathcal{F}}(w_0 h_\delta \circ f) = 2 \mathbb{E}_S \left[ \sup_{f \in \mathcal{F}} \left| \frac{1}{n_S/2} \sum_i \zeta_i w_0(X_i) h_\delta(Y_i^2 - f(X_i)) \right| \right]$$

$$= 2 \mathbb{E}_S \left[ \sup_{f \in \mathcal{F}} \left| \frac{1}{n_S/2} \sum_i \zeta_i \phi \left( w_0(X_i), Y_i^2 - f(X_i) \right) \right| \right] \quad [\phi(x, y) = x h_\delta(y)]$$

We first show that $\phi : \mathbb{R}^2 \to \mathbb{R}$ is a Lipschitz function on its domain. The first argument of $\phi$ is $w_0(x)$ which lies within $[-W, W]$. The second argument of $\phi$ is $Y^2 - f(X)$ (on the source domain), which is bounded by $B$. Therefore, $h_\delta(Y^2 - f(X))$ is bounded above by $(B+\delta)/\delta$. The derivative of $h_\delta$ is 0 for $x \leq -\delta$ and $\delta$ for $x \geq -\delta$. Hence, we have the following:

$$\|\nabla \phi(x, y)\| = \|(h_\delta(y) \quad x h_\delta'(y))\| \leq \sqrt{\frac{(B+\delta)^2}{\delta^2} + \frac{W^2}{\delta^2}} \leq \frac{B + W + \delta}{\delta} \, .$$

We next apply vector-valued Ledoux-Talagrand contraction inequality on the function $\phi$ (equation (1) of [Mau16]), to obtain the following bound on the Rademacher complexity:

$$2 \mathbb{E}_S \left[ \sup_{f \in \mathcal{F}} \left| \frac{1}{n_S/2} \sum_i \zeta_i \phi \left( w_0(X_i), Y_i^2 - f(X_i) \right) \right| \right]$$

$$\leq 2\sqrt{2} \left( \frac{B + W + \delta}{\delta} \right) \mathbb{E}_S \left[ \sup_{f \in \mathcal{F}} \left| \frac{1}{n_S/2} \sum_i \left( \zeta_{i1} w_0(X_i) + \zeta_{i2}(Y_i^2 - f(X_i)) \right) \right| \right]$$

$$\leq 2\sqrt{2} \left( \frac{B + W + \delta}{\delta} \right) \left[ \mathbb{E}_S \left[ \left\| \frac{1}{n_S/2} \sum_i \zeta_{i1} w_0(X_i) \right\| \right] + \mathbb{E}_S \left[ \left\| \frac{1}{n_S/2} \sum_{i \in \mathcal{D}_{S,1}} \zeta_{i,2} Y_i^2 \right\| \right] \mathcal{R}_{n_S/2}(\mathcal{F}) \right]$$

$$\leq 2\sqrt{2} \left( \frac{B + W + \delta}{\delta} \right) \left[ \frac{\|w_0\|_{L_2(P_{X_S})}}{\sqrt{n_S/2}} + \sqrt{\frac{\mathbb{E}_S[Y^4]}{n_S/2}} + \mathcal{R}_{n_S/2}(\mathcal{F}) \right]$$

Using this, we obtain the following:

$$\mathbb{E}_T\left(h_\delta\left(Y^2 - \hat{f}_{\text{init}}(X)\right)\right)$$

$$\leq \epsilon + \frac{B+\delta}{\delta}\left(\mathbb{E}_S\left[|\hat{w}(X) - w_0(X)|\right] + (2W+W')\sqrt{\frac{5\log(n_S/2)}{n_S/2}}\right)$$

$$+ 4\sqrt{2}\left(\frac{B+W+\delta}{\delta}\right)\left[\frac{\|w_0\|_{L_2(P_{X_S})} + \sqrt{\mathbb{E}_S[Y^4]}}{\sqrt{n_S}} + \mathcal{R}_{n_S/2}(\mathcal{F})\right]$$

$$\leq \epsilon + 4\sqrt{2}\left(\frac{B+W+\delta}{\delta}\right)\left[\mathbb{E}\left[|\hat{w}(X_S) - w(X_S)|\right] + (2W+W')\sqrt{\frac{5\log(n_S/2)}{n_S/2}}\right.$$

$$\left. + \frac{\|w_0\|_{L_2(P_{X_S})} + \sqrt{\mathbb{E}_S[Y^4]}}{\sqrt{n_S/2}} + \mathcal{R}_{n_S/2}(\mathcal{F})\right]$$

$$\leq \epsilon + 4\sqrt{2}\left(\frac{B+W+\delta}{\delta}\right)\left[\mathbb{E}\left[|\hat{w}(X_S) - w(X_S)|\right] + (2W+W')\sqrt{\frac{5\log(n_S/2)}{n_S/2}} + \frac{W + \sqrt{\mathbb{E}_S[Y^4]}}{\sqrt{n_S/2}} + \mathcal{R}_{n_S/2}(\mathcal{F})\right]$$

Choosing

$$\epsilon = L\delta + W\sqrt{\frac{5\log(n_S/2)}{n_S/2}} + \frac{B+\delta}{\delta}\cdot\left(\mathbb{E}_S\left[|\hat{w}(X) - w_0(X)|\right] + (W+W')\sqrt{\frac{5\log(n_S/2)}{n_S/2}}\right),$$

we obtain

$$\mathbb{E}_T\left(h_\delta\left(Y^2 - \hat{f}_{\text{init}}(X)\right)\right)$$

$$\lesssim L\delta + \frac{B+W+\delta}{\delta}\cdot\left(\left(\mathbb{E}_S\left[|\hat{w}(X) - w_0(X)|\right] + (W+W')\sqrt{\frac{5\log n_S}{n_S}} + \mathcal{R}_{n_S/2}(\mathcal{F})\right)\right)$$

$$\lesssim \sqrt{L(B+W)\left(\left(\mathbb{E}_S\left[|\hat{w}(X) - w_0(X)|\right] + (W+W')\sqrt{\frac{5\log n_S}{n_S}} + \mathcal{R}_{n_S/2}(\mathcal{F})\right)\right)}$$

$$+ \left(\mathbb{E}_S\left[|\hat{w}(X) - w_0(X)|\right] + (W+W')\sqrt{\frac{5\log n_S}{n_S}} + \mathcal{R}_{n_S/2}(\mathcal{F})\right)$$

$$\text{(by choosing } \delta \text{ to balance the terms)}$$

$$\triangleq \tau$$

Call the above event $\Omega_1$. This completes the proof of Step 1.

**Step 2:** Coming back to $\mathcal{D}_{S,2}$, we have:

$$\frac{1}{n_S/2}\sum_{i\in\mathcal{D}_{S,2}}\hat{w}(X_{S,i})\mathbb{1}_{Y_i^2 > \hat{f}_{\text{init}}(X_i)} \leq \frac{1}{n_S/2}\sum_{i\in\mathcal{D}_{S,2}}|\hat{w}(X_i) - w_0(X_i)| + \frac{1}{n_S/2}\sum_{i\in\mathcal{D}_{S,2}}w_0(X_i)\mathbb{1}_{Y_i^2 > \hat{f}_{\text{init}}(X_i)}$$

Furthermore, by Hoeffding's inequality, we have with probability $\geq 1 - e^{-t}$:

$$\frac{1}{n_S/2}\sum_{i\in\mathcal{D}_2}w_0(X_i)\mathbb{1}_{Y_i^2 > \hat{f}_{\text{init}}(X_i)} \leq \mathbb{E}_S\left[w_0(X)\mathbb{1}_{Y^2 > \hat{f}_{\text{init}}(X)}\right] + W\sqrt{\frac{t}{n_S}}$$

$$\leq \mathbb{E}_S\left[w_0(X)h_\delta\left(Y^2 - \hat{f}_{\text{init}}(X)\right)\right] + W\sqrt{\frac{t}{n_S}}$$

$$= \mathbb{E}_T\left(h_\delta\left(Y^2 - \hat{f}_{\text{init}}(X)\right)\right) + W\sqrt{\frac{t}{n_S}}$$

Meanwhile, with probability $\geq 1 - e^{-t}$:

$$\frac{1}{n_S/2}\sum_{i\in\mathcal{D}_{S,2}}|\hat{w}(X_i) - w_0(X_i)| \leq \mathbb{E}_S\left[|\hat{w}(X) - w_0(X)|\right] + (W+W')\sqrt{\frac{t}{n_S}}.$$

Therefore, with $t = 10 \log n_S$, we have with probability $\geq 1 - 2n_S^{-10}$:

$$\frac{1}{n_S/2} \sum_{i \in \mathcal{D}_{S,2}} \hat{w}(X_i) \mathbb{1}_{Y_i^2 > \hat{f}_{\text{init}}(X_i)} \leq \mathbb{E}_S \left[ |\hat{w}(X) - w_0(X)| \right] + (W + W') \sqrt{\frac{10 \log n_S}{n_S}}$$

$$+ \mathbb{E}_T \left( h_\delta \left( Y^2 - \hat{f}_{\text{init}}(X) \right) \right) + W \sqrt{\frac{10 \log n_S}{n_S}} \, .$$

Call this event $\Omega_2$. Therefore, on $\Omega_1 \cap \Omega_2$ we have:

$$\frac{1}{n_S/2} \sum_{i \in \mathcal{D}_{S,2}} \hat{w}(X_i) \mathbb{1}_{Y_i^2 > \hat{f}_{\text{init}}(X_i)} \leq \mathbb{E}_S \left[ |\hat{w}(X) - w_0(X)| \right] + (2W + W') \sqrt{\frac{10 \log n_S}{n_S}} + \tau \triangleq \tilde{\tau} \, .$$

This completes the proof of Step 2. For any fixed $\alpha > 0$, we have $\tilde{\tau} \leq \alpha$ as long as $n_S$ is large enough and $\mathbb{E}_S \left[ |\hat{w}(X) - w_0(X)| \right]$ is small enough, and as a consequence $\hat{\lambda}(\alpha) \leq 1$. This completes the proof.

### A.3 Proof of Theorem 3.4

Recall that we construct the prediction intervals using data splitting; from the first part of the data (namely $\mathcal{D}_1$), we estimate $\hat{f}_{\text{init}}$ and use the second part of the data (namely $\mathcal{D}_2$) to estimate $\hat{\lambda}(\alpha)$. Conditional on $\mathcal{D}_1$, define a function class $\mathcal{G} \equiv \mathcal{G}(\hat{f})$ as:

$$\mathcal{G} = \left\{ g_\lambda(x, y) = w_0(x) \mathbb{1}_{y^2 - \lambda \hat{f}_{\text{init}}(x) \geq 0} : \lambda \geq 0 \right\} \, .$$

As $\mathcal{G}$ only depends on a scalar parameter $\lambda$ (as $w_0$ and $\hat{f}_{\text{init}}$ are fixed conditionally on $\mathcal{D}_{S,1}, \mathcal{D}_T$), it is a VC class of function with VC-dim $\leq 2$.

$$\mathbb{P}_T \left( Y^2 \geq \hat{\lambda}(\alpha) \hat{f}_{\text{init}}(X) \right)$$

$$= \mathbb{E}_S \left[ w_0(X) \mathbb{1}_{Y^2 - \hat{\lambda}(\alpha) \hat{f}_{\text{init}}(X) \geq 0} \right]$$

$$= \frac{1}{n_S/2} \sum_{i \in \mathcal{D}_{S,2}} w_0(X_i) \mathbb{1}_{Y_i^2 - \hat{\lambda}(\alpha) \hat{f}_{\text{init}}(X_i)} + (\mathbb{P}_S - \mathbb{P}_{n_S/2}) w_0(X) \mathbb{1}_{Y^2 \geq \hat{\lambda}(\alpha) \hat{f}_{\text{init}}(X)}$$

$$= \frac{1}{n_S/2} \sum_{i \in \mathcal{D}_{S,2}} \hat{w}(X_i) \mathbb{1}_{Y_i^2 - \hat{\lambda}(\alpha) \hat{f}_{\text{init}}(X_i) \geq 0} + \frac{1}{n_S/2} \sum_{i \in \mathcal{D}_{S,2}} (w_0(X_i) - \hat{w}(X_i)) \mathbb{1}_{Y_i^2 - \hat{\lambda}(\alpha) \hat{f}_{\text{init}}(X_i) \geq 0}$$

$$+ (\mathbb{P}_S - \mathbb{P}_{n_S/2}) w_0(X) \mathbb{1}_{Y^2 - \hat{\lambda}(\alpha) \hat{f}_{\text{init}}(X) \geq 0} \tag{A.1}$$

Now, by the definition of $\hat{\lambda}(\alpha)$ (see Step 2), we have:

$$\alpha - \frac{1}{n_S/2} \leq \frac{1}{n_S/2} \sum_{i \in \mathcal{D}_{S,2}} \hat{w}(X_i) \mathbb{1}_{Y_i^2 - \hat{\lambda}(\alpha) \hat{f}_{\text{init}}(X_i) \geq 0} \leq \alpha \, .$$

We use a similar technique to control the second summand as in the proof of Theorem 3.2. By using the fact that the indicator function is less than one, we have:

$$\left| \frac{1}{n_S/2} \sum_{i \in \mathcal{D}_{S,2}} (w_0(X_i) - \hat{w}(X_i)) \mathbb{1}_{Y_i^2 - \hat{\lambda}(\alpha) \hat{f}_{\text{init}}(X_i) \geq 0} \right| \leq \frac{1}{n_S/2} \sum_{i \in \mathcal{D}_{S,2}} |\hat{w}(X_i) - w_0(X_i)| \, .$$

Applying Hoeffding's inequality (with the fact that $\|\hat{w}\|_\infty \leq W'$ and $\|w_0\|_\infty \leq W$), we have with probability greater than or equal to $1 - e^{-t}$:

$$\frac{1}{n_S/2} \sum_{i \in \mathcal{D}_{S,2}} |\hat{w}(X_i) - w_0(X_i)| \leq \mathbb{E}_S \left[ |\hat{w}(X) - w(X)| \right] + (W + W') \sqrt{\frac{t}{n_S}} \, .$$

To control the third summand of (A.1), note that, conditional on $\mathcal{D}_{S,1}$ and $\mathcal{D}_T$ (i.e., assuming $\hat{f}_{\text{init}}$ fixed), and using the fact that $\|g\|_\infty \leq \|w_0\|_\infty \leq W$ for all $g \in \mathcal{G}$, we have by Mcdiarmid's

inequality with probability greater than or equal to $1 - e^{-t}$:

$$\sup_{g \in \mathcal{G}} \left| (\mathbb{P}_S - \mathbb{P}_{n_S/2}) g(X, Y) \right| \leq \mathbb{E}_S \left[ \sup_{g \in \mathcal{G}} \left| (\mathbb{P}_S - \mathbb{P}_{n_S/2}) g(X, Y) \right| \mid \mathcal{D}_{S,1}, \mathcal{D}_T \right] + W \sqrt{\frac{t}{n_S}}$$

$$\leq 2 \mathcal{R}_{n_S/2} \left( \mathcal{G} \mid \mathcal{D}_{S,1}, \mathcal{D}_T \right) + W \sqrt{\frac{t}{n_S}}.$$

Now conditional on $\mathcal{D}_{S,1}, \mathcal{D}_T$, $\mathcal{G}$ is a VC class of function with VC dimension $\leq 2$. Therefore,

$$\mathcal{R}_{n_S/2} \left( \mathcal{G} \mid \mathcal{D}_{S,1}, \mathcal{D}_T \right) \leq \sqrt{\frac{C}{n_S}}$$

for some constant $C > 0$. Thus, we have

$$\sup_{g \in \mathcal{G}} \left| (\mathbb{P}_S - \mathbb{P}_{n_S/2}) g(X, Y) \right| \leq \sqrt{\frac{C}{n_S}} + W \sqrt{\frac{t}{n_S}}.$$

Combining the bounds, we have, with probability $\geq 1 - 2e^{-t}$:

$$\left| \mathbb{P}_T \left( Y^2 > \hat{\lambda}(\alpha) \hat{f}_{\text{init}}(X) \right) - \alpha \right|$$

$$\leq \frac{1}{n_S/2} + \mathbb{E}_S \left[ |\hat{w}(X) - w_0(X)| \right] + (2W + W') \sqrt{\frac{t}{n_S}} + \sqrt{\frac{C}{n_S}}.$$

This completes the proof.

### A.4 Proof of Theorem 4.3

We start with the following decomposition:

$$\mathbb{E}_T[\hat{f}_{\text{init}} \circ \hat{T}_0(X)] = \mathbb{E}_T[\hat{f}_{\text{init}} \circ T_0(X)] + \mathbb{E}_T[\hat{f}_{\text{init}} \circ \hat{T}_0(X) - \hat{f}_{\text{init}} \circ T_0(X)]$$

$$= \mathbb{E}_S[\hat{f}_{\text{init}}(X)] + \mathbb{E}_T[\hat{f}_{\text{init}} \circ \hat{T}_0(X) - \hat{f}_{\text{init}} \circ T_0(X)]$$

$$\leq \mathbb{E}_S[\hat{f}_{\text{init}}(X)] + L_{\mathcal{F}} \mathbb{E}_T[|\hat{T}_0(X) - T_0(X)|]$$

where the second equation follows from the fact that when $X \sim P_T$, then $T_0(X) \sim P_S$, and the last line follows from the fact $f \in \mathcal{F}$ is $L_{\mathcal{F}}$ Lipschitz. A similar argument as in the proof of Theorem 3.5 [FGM23] yields:

$$\mathbb{E}_S[\hat{f}_{\text{init}}(X)] \leq \Delta + 4 \mathcal{R}_{n_S}(\mathcal{F}) + 4 B_{\mathcal{F}} \sqrt{\frac{t}{2n_S}}.$$

with probability $\geq 1 - e^{-t}$. We then finish the proofs.

### A.5 Proof of Lemma 4.4

By the definition of $\hat{\lambda}(\alpha)$, we have

$$\left\{ \hat{\lambda}(\alpha) \geq 1 \right\} \implies \left\{ \frac{1}{n_S/2} \sum_{i \in \mathcal{D}_{S,2}} \mathbb{1} \left( Y_i^2 \geq \hat{f}_{\text{init}}(X_i) + \delta \right) > \alpha \right\}.$$

Now by an application of Chernoff bound for binomial distribution, we have:

$$\mathbb{P} \left( \frac{1}{n_S/2} \sum_{i \in \mathcal{D}_{S,2}} \mathbb{1} \left( Y_i^2 \geq \hat{f}_{\text{init}}(X_i) + \delta \right) > \alpha \mid \mathcal{D}_{S,1}, \mathcal{D}_T \right) \leq e^{-\frac{(\alpha - p_{n_S})^2 n_S}{6 p_{n_S}}}.$$

Hence, we have the following:

$$\mathbb{P}(\hat{\lambda}(\alpha) > 1 \mid \mathcal{D}_{S,1}, \mathcal{D}_T) \leq e^{-\frac{(\alpha - p_{n_S})^2 n_S}{6 p_{n_S}}}.$$

We next establish the high probability bound on $p_{n_S}$. We define a function $\ell_\delta(x)$ which is 1 when $x \leq -\delta$, 0 when $x \geq 0$ and $-x/\delta$ when $-\delta \leq x \leq 0$.

$$
\begin{aligned}
p_{n_S} = \mathbb{E}_S\left[\mathbb{1}_{Y^2 \geq \hat{f}_{\text{init}}(X)+\delta}\right] &\leq \mathbb{E}_S\left[\ell_\delta(\hat{f}_{\text{init}}(X) - Y^2)\right] \\
&= \frac{1}{n_S/2} \sum_{i \in \mathcal{D}_{S,1}} \ell_\delta(\hat{f}_{\text{init}}(X_i) - Y_i^2) + \left(\mathbb{P}_{n_S/2} - \mathbb{P}_S\right) \ell_\delta(\hat{f}_{\text{init}}(X) - Y^2) \\
&\leq \sup_{f \in \mathcal{F}} \left(\mathbb{P}_{n_S/2} - \mathbb{P}_S\right) \ell_\delta(f(X) - Y^2) \\
&\leq \frac{4}{\delta}\left(\sqrt{\frac{\mathbb{E}_S[Y^4]}{n_S}} + \mathcal{R}_{n_S/2}(\mathcal{F})\right) + \sqrt{\frac{t}{n_S}}\,.
\end{aligned}
$$

where the first inequality used $\ell_\delta(x) \geq \mathbb{1}(x \leq -\delta)$, second inequality uses the fact that sample average of $\ell_\delta$ over $\mathcal{D}_{S,1}$ is 0 by the definition of $\hat{f}_{\text{init}}$, third inequality uses Ledoux-Talagrand contraction inequality observing that $\ell_\delta$ is $1/\delta$-Lipschitz. This completes the proof.

## A.6 Proof of Theorem 4.5

$$
\begin{aligned}
&\mathbb{P}_T\left(Y^2 \geq \hat{\lambda}(\alpha)(\hat{f}_{\text{init}} \circ \hat{T}_0(X) + \delta)\right) \\
&= \mathbb{P}_T\left(Y^2 \geq \hat{\lambda}(\alpha)(\hat{f}_{\text{init}} \circ T_0(X) + \delta)\right) \\
&\qquad + \left|\mathbb{P}_T\left(Y^2 \geq \hat{\lambda}(\alpha)(\hat{f}_{\text{init}} \circ \hat{T}_0(X) + \delta)\right) - \mathbb{P}_T\left(Y^2 \geq \hat{\lambda}(\alpha)(\hat{f}_{\text{init}} \circ T_0(X) + \delta)\right)\right| \\
&\triangleq T_1 + T_2\,.
\end{aligned} \tag{A.2}
$$

We start with analyzing the first term:

$$
\begin{aligned}
T_1 &= \mathbb{P}_T\left(Y^2 \geq \hat{\lambda}(\alpha)(\hat{f}_{\text{init}} \circ T_0(X) + \delta)\right) \\
&= \int_{\mathcal{X}_T} \int_{\mathcal{Y}} \mathbb{1}_{y^2 \geq \hat{\lambda}(\alpha)(\hat{f}_{\text{init}}(T_0(x))+\delta)}\, f_T(y \mid X_T = x)\, p_T(x)\, dydx \\
&= \int_{\mathcal{X}_T} \int_{\mathcal{Y}} \mathbb{1}_{y^2 \geq \hat{\lambda}(\alpha)(\hat{f}_{\text{init}}(T_0(x))+\delta)}\, f_S(y \mid X_S = T_0(x))\, p_T(x)\, dydx \\
&= \int_{\mathcal{X}_S} \int_{\mathcal{Y}} \mathbb{1}_{y^2 \geq \hat{\lambda}(\alpha)(\hat{f}_{\text{init}}(z)+\delta)}\, f_S(y \mid X_S = z)\, p_T(T_0^{-1}(z))|\nabla T_0^{-1}(z)|\, dydx \\
&= \int_{\mathcal{X}_S} \int_{\mathcal{Y}} \mathbb{1}_{y^2 \geq \hat{\lambda}(\alpha)(\hat{f}_{\text{init}}(z)+\delta)}\, f_S(y \mid X_S = z)\, p_S(z)\, dydx \\
&= \mathbb{P}_S(Y^2 \geq \hat{\lambda}(\alpha)(\hat{f}_{\text{init}}(X) + \delta))\,.
\end{aligned}
$$

Therefore, we need a high probability upper bound on $\mathbb{P}_S(Y^2 \geq \hat{\lambda}(\alpha)(\hat{f}_{\text{init}}(X) + \delta) \mid \mathcal{D}_S \cup \mathcal{D}_T)$. Towards that end, we start with the following expansion:

$$
\begin{aligned}
&\mathbb{P}_S\left(Y^2 \geq \hat{\lambda}(\alpha)(\hat{f}_{\text{init}}(X) + \delta) \mid \mathcal{D}_S \cup \mathcal{D}_T\right) \\
&= \frac{1}{n_S/2} \sum_{i \in \mathcal{D}_{S,2}} \mathbb{1}_{Y_i^2 \geq \hat{\lambda}(\alpha)(\hat{f}_{\text{init}}(X_i)+\delta)} + \left(\mathbb{P}_{n_S/2} - \mathbb{P}_S\right) \mathbb{1}_{Y^2 \geq \hat{\lambda}(\alpha)(\hat{f}_{\text{init}}(X)+\delta)}
\end{aligned} \tag{A.3}
$$

Now, note that, by the definition of $\hat{\lambda}(\alpha)$, we have:

$$
\alpha - \frac{1}{n_S/2} \leq \frac{1}{n_S/2} \sum_{i \in \mathcal{D}_{S,2}} \mathbb{1}_{Y_i^2 \geq \hat{\lambda}(\alpha)(\hat{f}_{\text{init}}(X_i)+\delta)} \leq \alpha\,.
$$

To bound the second term in (A.3), we use:

$$
\left|\left(\mathbb{P}_{n_S/2} - \mathbb{P}_S\right) \mathbb{1}_{Y^2 \geq \hat{\lambda}(\alpha)(\hat{f}_{\text{init}}(X)+\delta)}\right| \leq \sup_{\lambda \geq 0} \left|\left(\mathbb{P}_{n_S/2} - \mathbb{P}_S\right) \mathbb{1}_{Y^2 \geq \lambda(\hat{f}_{\text{init}}(X)+\delta)}\right| := \mathbf{Z}_n\,.
$$

To bound the supremum we use standard techniques from the empirical process theory. Define a collection of functions $\mathcal{G} = \left\{ \mathbb{1}_{Y^2 \geq \lambda(\hat{f}_{\mathrm{init}}(X) + \delta)} : \lambda \geq 0 \right\}$. Note that, here we condition on $\mathcal{D}_{S,1}$, so we treat $\hat{f}_{\mathrm{init}}$ as a constant function. For notational simplicity, suppose

$$\Psi_n = \mathbb{E}_S \left[ \sup_{\lambda \geq 0} \left| (\mathbb{P}_{n_S/2} - \mathbb{P}_S) \, \mathbb{1}_{Y^2 \geq \lambda(\hat{f}_{\mathrm{init}}(X) + \delta)} \right| \mid \mathcal{D}_{S,1} \right] = \mathbb{E}_S \left[ \sup_{g \in \mathcal{G}} \left| (\mathbb{P}_{n_S/2} - \mathbb{P}_S) \, g(X, Y) \right| \mid \mathcal{D}_{S,1} \right].$$

As the functions in $\mathcal{G}$ are uniformly bounded by 1 (and consequently, $\mathbb{E}[g^2(X, Y)] \leq 1$), we have by Talagrand's concentration inequality of the suprema of the empirical process:

$$\mathbb{P} \left( \mathbf{Z}_n \geq \Psi_n + \sqrt{2t \frac{1 + 4\Psi_n}{n_S}} + \frac{4t}{3n_S} \mid \mathcal{D}_{S,1} \right) \leq e^{-t}. \tag{A.4}$$

Therefore, we need an upper bound on $\Psi_n$ to obtain a high probability upper bound on $\mathbf{Z}_n$. Towards that end, observe that $\mathcal{G}$ is a VC class with VC-dim less than or equal to 2 (as it is an indicator function of a collection of functions with one parameter). Hence, we have, by symmetrization and Dudley's metric entropy bound:

$$\Psi_n \leq 2\mathbb{E}_S \left[ \sup_{g \in \mathcal{G}} \left| \frac{1}{n_S/2} \sum_{i \in \mathcal{D}_{S,2}} \epsilon_i g(X_i, Y_i) \right| \mid \mathcal{D}_{S,1} \right] \leq \frac{C}{\sqrt{n_S}}.$$

Therefore, going back to (A.4), we have with probability $\geq 1 - e^{-t}$

$$\mathbf{Z}_n \leq \frac{C}{\sqrt{n_S}} + \sqrt{\frac{C_1}{n_S} + \frac{C_2}{n_S^{3/2}}} \sqrt{t} + \frac{4t}{3n_S}.$$

Hence, we have:

$$\left| \mathbb{P}_S \left( Y^2 \geq \hat{\lambda}(\alpha)(\hat{f}_{\mathrm{init}}(X) + \delta) \mid \mathcal{D}_S \cup \mathcal{D}_T \right) - \alpha \right| \lesssim \sqrt{\frac{t}{n_S}}$$

with probability $\geq 1 - e^{-t}$. This completes the proof of $T_1$. To obtain a bound on $T_2$, note that:

$$T_2$$
$$= \left| \mathbb{P}_T \left( Y^2 \geq \hat{\lambda}(\alpha)(\hat{f}_{\mathrm{init}} \circ \hat{T}_0(X) + \delta) \right) - \mathbb{P}_T \left( Y^2 \geq \hat{\lambda}(\alpha)(\hat{f}_{\mathrm{init}} \circ T_0(X) + \delta) \right) \right|$$
$$= \left| \int_{\mathcal{X}_T} \left( \mathbb{P}_T(Y^2 \leq \hat{\lambda}(\alpha)(\hat{f}_{\mathrm{init}}(\hat{T}_0(x)) + \delta) \mid X_T = x) \right. \right.$$
$$\left. \left. - \mathbb{P}_T(Y^2 \leq \hat{\lambda}(\alpha)(\hat{f}_{\mathrm{init}}(T_0(x)) + \delta) \mid X_T = x) \right) p_T(x) \, dx \right|$$
$$= \left| \int_{\mathcal{X}_T} \left( F_{Y_T^2 \mid X_T = x}(\hat{\lambda}(\alpha)(\hat{f}_{\mathrm{init}}(\hat{T}_0(x)) + \delta)) - F_{Y_T^2 \mid X_T = x}(\hat{\lambda}(\alpha)(\hat{f}_{\mathrm{init}}(T_0(x)) + \delta)) \right) p_T(x) \, dx \right|$$
$$\leq G \int_{\mathcal{X}_T} \hat{\lambda}(\alpha) \left| \hat{f}_{\mathrm{init}}(T_0(x)) - \hat{f}_{\mathrm{init}}(\hat{T}_0(x)) \right| p_T(x) \, dx$$
$$\leq G L_{\mathcal{F}} \, \mathbb{E}_T[|T_0(X) - \hat{T}_0(X)|].$$

Here, the penultimate inequality uses the fact that the conditional distribution of $Y_T^2$ given $X_T$ is Lipschitz (as the density of $Y_T^2$ given $X_T$ is bounded), and the last inequality uses the fact that $\hat{f}_{\mathrm{init}}$ is Lipschitz as we have assumed all functions in $\mathcal{F}$ are Lipschitz.

## B  Details of the experiment

### B.1  Density ratio estimation via probabilistic classification

Suppose we observe $\{X_1, \ldots, X_{n_1}\}$ from a distribution $P$ (with density $p$) and $\{X_{n_1+1}, \ldots, X_{n_1+n_2}\}$ from another distribution $Q$ (with density $q$). We are interested in estimating $w_0(x) = q(x)/p(x)$, where we assume $Q$ is absolutely continuous with respect to $P$

(otherwise, the density ratio can be unbounded with positive probability). Define, $n_1 + n_2$ mane binary random variables $\{C_1, \ldots, C_{n_1+n_2}\}$ such that $C_i = 0$ for $1 \leq i \leq n_1$ and $C_i = 1$ for $n_1 + 1 \leq i \leq n_1 + n_2$. Consider the augmented dataset $\mathcal{D} = \{(X_i, C_i)\}_{1 \leq i \leq n_1+n_2}$. We can think that this dataset is generated from a mixture distribution $\rho p(X) + (1-\rho)q(x)$ where $\rho = \mathbb{P}(C = 1)$. For this mixture distribution, the posterior distribution of $C$ given $X$ is:

$$\mathbb{P}(C = 1 | X = x) = \frac{P(X = x \mid C = 1)P(C = 1)}{P(X = x \mid C = 1)P(C = 1) + P(X = x \mid C = 0)P(C = 0)}$$
$$= \frac{\rho q(x)}{\rho q(x) + (1-\rho)p(x)}$$
$$= \frac{(\rho/(1-\rho))w_0(x)}{(\rho/(1-\rho))w_0(x) + 1}$$

This implies:

$$w_0(x) = \frac{1-\rho}{\rho} \frac{\mathbb{P}(C = 1 \mid X = x)}{1 - \mathbb{P}(C = 1 \mid X = x)} .$$

Now, from the data, we can estimate $\hat{\rho} = n_2/(n_1+n_2)$ and $\mathbb{P}(C = 1 \mid X = x)$ by any classification technique (e.g., using logistic regression, boosting, random forest, deep neural networks etc). Let $\hat{g}(x)$ be one such classifier. Then we can estimate $w_0(x)$ by $(n_1/n_2)(\hat{g}(x)/(1 - \hat{g}(x)))$.

## B.2 General weighted conformal prediction

The weighted conformal prediction method, as presented in [TFBCR19], consists of two main steps:

1. Split the source data into parts; estimate the conditional mean function $\mathbb{E}[Y \mid X = x]$, say $\hat{\mu}(x)$ using the first part of the source data.

2. Use the second part of the source data and the target data to construct weight $w(X_i)$ and the score function $S(x, y) = |y - \hat{\mu}(x)|$ to construct the confidence interval.

In Section 5, we have implemented a generalized version of it, where we modify the score function as follows:

1. We estimate the conditional standard deviation function $\sqrt{\mathrm{var}(Y \mid X = x)}$ along with the conditional mean function from the first part of the data. Call it $\hat{\sigma}(x)$.

2. We use the modified score function $s(x, y) = |y - \hat{\mu}(x)|/\hat{\sigma}(x)$.

The rest of the method is the same as [TFBCR19]. This additional estimated conditional variance function allows more expressivity and flexibility to the conformal prediction band, as observed in Section 5.2 of [LGR$^+$18], as this captures the local heterogeneity of the conditional distribution of $Y$ given $X$.

## B.3 Boxplots to compare coverage and bandwidth

In this subsection, we present two boxplots to compare the variation in coverage and average width of the prediction bands between our method and the generalized weighted conformal prediction (as described in the previous subsection).

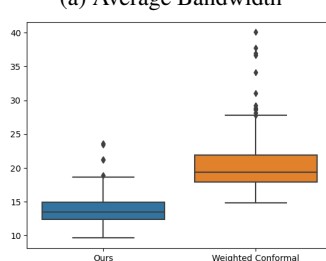
(a) Average Bandwidth

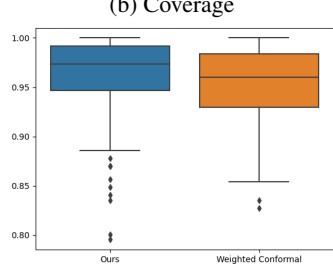
(b) Coverage

| Real Estate Data | | | |
| --- | --- | --- | --- |
| Outcome | Our Method | WVAC | WQC |
| Coverage (Median) | 0.98 | 0.962 | 0.971 |
| Coverage (IQR) | 0.058 | 0.048 | 0.048 |
| Bandwidth (Median) | 36.889 | 46.392 | 46.858 |
| Bandwidth (IQR) | 15.001 | 19.027 | 14.189 |
| Bandwidth (Median for Coverage > 95%) | 40.774 | 50.703 | 51.031 |

Table 2: Experimental results for the real estate data

The boxplots immediately show that our methods yield similar coverage (even with lesser variability) with significantly lower average width than the generalized weighted conformal prediction method.

## C    Additional Experiments

In this section, we present the details of the additional experiments based on four other datasets, as mentioned in Section 5. We compare our methodology against two other widely used conformal methods, namely Weighted Variance-adjusted Conformal (WVAC) [TFBCR19], and Weighted Quantile-adjusted Conformal (WQC)—which is a variant of WVAC, where the score function is changed to the conditional quantile [RPC19].

### C.1    Experiment 1: Real estate data

The Real Estate Valuation dataset available at the UCI Machine Learning Repository contains data used to estimate the value of real estate properties. Collected from Taipei, Taiwan, this dataset is useful for predictive modeling in the real estate sector. It consists of 414 instances, with each entry representing a different real estate transaction. This dataset was originally introduced in [TX12a]. In this dataset, the goal is to predict the house price per unit area based on the 6 other features, namely i) transaction date, ii) house age, iii) distance to the nearest MRT station, iv) number of convenience stores, v) latitude and vi) longitude. The construction of shifted data (with $\beta = (-1, 0, -1, 0, 1, 1)$) and implementation procedure are the same as in Section 5. Table 2 and Figure 3 presented the results over 200 Monte Carlo iterations. It is evident from the table that our method produced a small average width in comparison to the other methods while maintaining the coverage guarantee.

### C.2    Experiment 2: Energy efficiency data

The Energy Efficiency dataset available at the UCI Machine Learning Repository is designed to help predict the heating load and cooling load requirements of buildings. The dataset was originally introduced in [YH18]. The dataset includes various building parameters such as: i) wall area, ii) surface area, iii) roof area, iv) orientation, etc. The goal is to predict the heating load based on 8 other covariates. The construction of shifted data (with $\beta = (-1, 0, 1, 0, -1, 0, 0, -1)$) and implementation procedure are the same as in Section 5. Our results over 200 Monte Carlo iterations are presented in Table 3 and Figure 4. The table shows that our method produced a smaller bandwidth than WVAC. While WQC has a smaller median bandwidth, it sacrifices coverage. The last row indicates that for experiments with coverage $\geq 95\%$, WQC's median average width is significantly larger than ours. Thus, whenever WQC provides adequate coverage, its bandwidth is much larger than ours.

### C.3    Experiment 3: Appliances Energy Prediction Dataset

Appliances Energy Prediction Dataset is freely available from the UCI repository. This dataset is a time series data with 28 covariates and one response variable. We used data from 2016-01-11 to 2016-02-15 (5000 samples) as our training set and data from 2016-05-13 to 2016-05-27 (2000

| Energy efficiency data | | | |
|---|---|---|---|
| Outcome | Our Method | WVAC | WQC |
| Coverage (Median) | 0.995 | 0.969 | 0.973 |
| Coverage (IQR) | 0.047 | 0.036 | 0.05 |
| Bandwidth (Median) | 4.332 | 5.045 | 2.842 |
| Bandwidth (IQR) | 1.358 | 3.269 | 2.551 |
| Bandwidth (Median for Coverage > 95%) | 4.373 | 5.681 | 4.94 |

Table 3: Experimental results for the energy efficiency data

samples) as our testing set. Since the source and target data are from different time periods, this experiment involves a non-synthetic real-world time shift. The results based on our Algorithm 2 are presented below:

| Appliances Energy Prediction Dataset | | | |
|---|---|---|---|
| Outcome | Our Method | WVAC | WQC |
| Coverage | 0.95 | 1.00 | 1.00 |
| Bandwidth | 461.69 | 6809.87 | 2032.12 |

Table 4: Experimental results for the Appliances Energy Prediction Dataset

## C.4 Experiment 4: ETDataset

We applied our method to the ETDataset (ETT-small) from [ZZP+21], which contains hourly-level data from two electricity transformers at two different stations, including load and oil temperature measurements. Each data point consists of 8 features, including the date of the point, the predictive value "oil temperature", and 6 different types of external power load features. For our experiment, we used the data from one transformer during the period from July 1, 2016, to November 2, 2016, as our source data, and data from the same time period from the other transformer as our target data. As our source data and the target data are from different locations, we have a geographical covariate shift; see [ZZP+21] for more details. Our results are as follows:

| ETDataset | | | |
|---|---|---|---|
| Outcome | Our Method | WVAC | WQC |
| Coverage | 0.976 | 0.982 | 0.842 |
| Bandwidth | 41.525 | 57.9 | 54.981 |

Table 5: Experimental results for the ETDataset

## C.5 Experiment 5: Airfoil data

In Section 5, we have implemented our Algorithm 1 on the Airfoil data [TFBCR19] to showcase the efficacy of our method. Here, additionally, we implement Algorithm 2 on the same dataset to evaluate the performance of our second method based on optimal transport-based domain alignment. Here, the data shifting procedure (to create data from the target domain with a shifted distribution) is different from the rest of the experiments; we use the linear transformation $x \mapsto Ax + b$ with $A = \text{diag}(1.5, 1.2, 1.6, 2, 1.8)$ and $b = (1, 0, 0, 1, 0)$ to generate covariates on the target domain. As before, we split the data into two parts; we keep 75% of the data as it is, and shift the rest 25% of the data. The results are given in Table 6 and Figure 5. The second column of Table 6, namely *Our Method (Without OT)* is the aggregation method proposed in this paper, without domain alignment

| Airfoil data | | | | |
|---|---|---|---|---|
| Outcome | Our Method | Our Method (without OT) | WVAC | WQC |
| Coverage (Median) | 0.928 | 0.749 | 0.984 | 0.952 |
| Coverage (IQR) | 0.035 | 0.22 | 0.024 | 0.077 |
| Bandwidth (Median) | 15.075 | 18.512 | 36.298 | 32.143 |
| Bandwidth (IQR) | 1.638 | 3.089 | 10.619 | 8.364 |
| Bandwidth (Median for Coverage > 95%) | 16.429 | 25.268 | 37.783 | 36.433 |

Table 6: Experimental results for the airfoil data using optimal transport

through optimal transport, i.e., we use source data to construct the prediction interval and use the same prediction interval on the target domain. As evident from Table 6, our method outperforms all other methods in terms of the average width of the prediction interval while maintaining a good coverage guarantee.

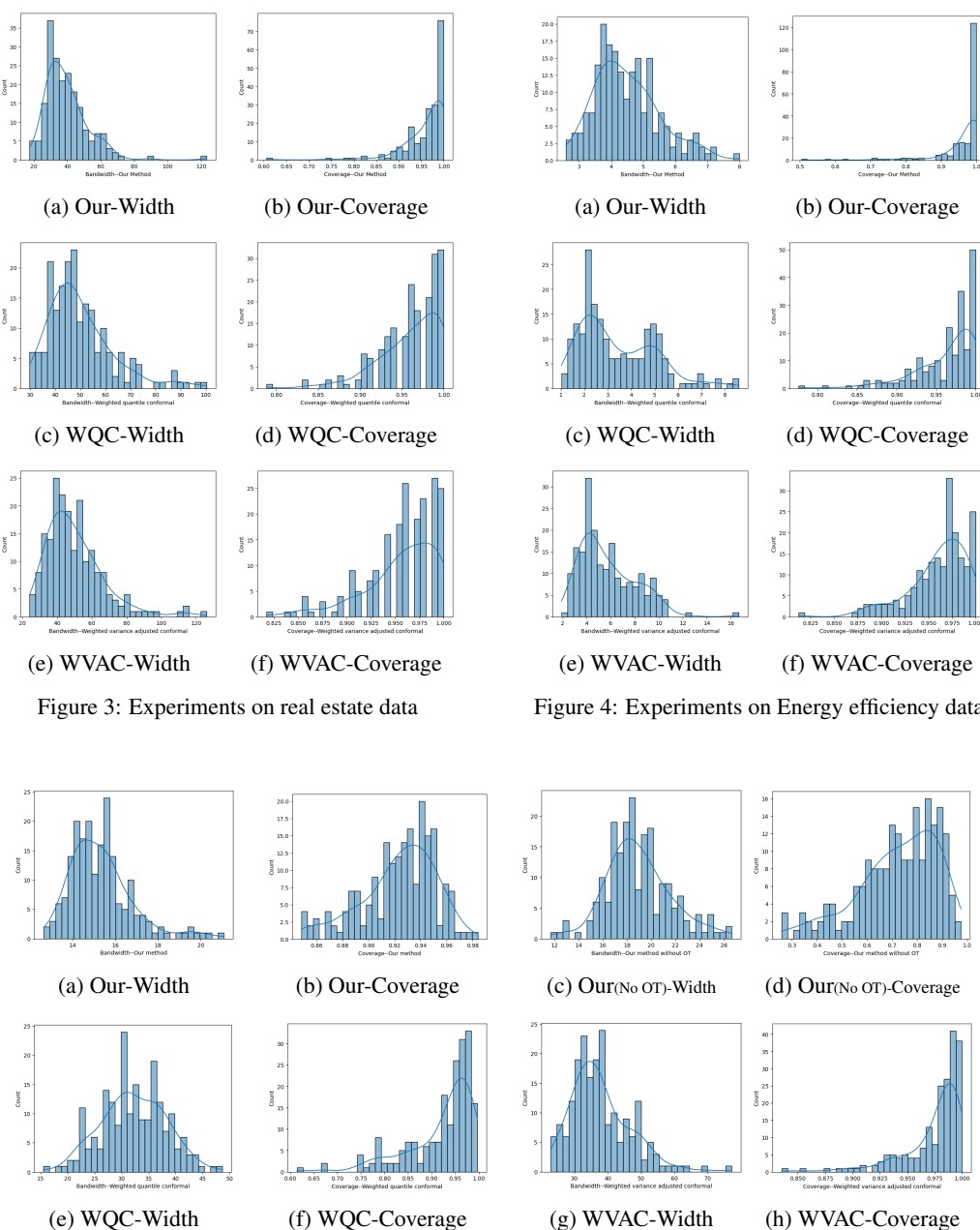

(a) Our-Width   (b) Our-Coverage   (a) Our-Width   (b) Our-Coverage

(c) WQC-Width   (d) WQC-Coverage   (c) WQC-Width   (d) WQC-Coverage

(e) WVAC-Width   (f) WVAC-Coverage   (e) WVAC-Width   (f) WVAC-Coverage

Figure 3: Experiments on real estate data   Figure 4: Experiments on Energy efficiency data

(a) Our-Width   (b) Our-Coverage   (c) Our(No OT)-Width   (d) Our(No OT)-Coverage

(e) WQC-Width   (f) WQC-Coverage   (g) WVAC-Width   (h) WVAC-Coverage

Figure 5: Experiments on Airfoil data using optimal transport

