# OpenReview forum: "Optimal Aggregation of Prediction Intervals under Unsupervised Domain Shift"
_NeurIPS.cc/2024/Conference — NeurIPS 2024 poster_

### Official Review · Reviewer_EZSx · 2024-07-06

**Soundness:** 3
**Presentation:** 3
**Contribution:** 3
**Rating:** 6
**Confidence:** 2

**Summary:**

It is crucial to assess and quantify uncertainties associated with distribution shifts in the context of unsupervised domain shift. The paper propose methodologies for aggregating prediction intervals, which capture the range of likely outcomes for a given prediction. The authors support these methodologies theoretically by considering a bounded density ratio and a measure-preserving transformation between source and target domains. This includes finite sample bounds for the coverage and width of their prediction intervals.

**Strengths:**

1.The method is innovative in aggregating various prediction intervals on the target domain under the domain-aligning assumption within an unsupervised domain adaptation framework.

2.The paper includes sufficient theoretical discussions, presenting finite sample concentration bounds to demonstrate that their method achieves adequate coverage with a small width.

**Weaknesses:**

1.Apart from theoretical results, there is a lack of intuitive understanding as to why aggregation can achieve minimal-width prediction intervals.

2.The paper does not include comparisons with previous methods of aggregated prediction intervals (Hosen et al., 2014).

3.Limited experiments and comparative methods make it difficult to affirm the effectiveness of the algorithm.

[1] Hosen, A., Khosravi, A., Nahavandi, S., and Creighton, D. Improving the Quality of Prediction Intervals Through Optimal Aggregation. IEEE Transactions on Industrial Electronics. 2014

**Questions:**

1.What does $f \in \mathcal{F}$ signify in equation (2.1)? There is no $f$ explicitly used in the main part of this equation.

2.What does $m_0$ represent in line 118? Is it the same as $m_0$ mentioned in line 137? What is the relationship between $m_0$ and $f_0$?

3.In line 164, it states, "However, any such estimator $ \hat{\omega}(x) $ can be non-zero for $ x $ where $ \omega_0(x)=0 $ due to estimation error. Consequently, $ \hat{\omega} $ may not be efficient in selecting informative source samples." Given this observation, why not apply the hinge function not to $ \hat{\omega} $, but to $ (Y - m_0(x))^2 - f(x) $ in Equation (3.5)?

4.The notation $B_{\mathcal{F}}$ first appears in Theorem 3.2 (please correct me if I am mistaken), and there is no explanation provided for it.

5.The paper introduces two algorithms, but Section 5 includes experiments for only one of them. Which algorithm is it, and why are there no experiments conducted on the other one?

---

> ### Author Rebuttal · Authors · 2024-08-06
>
> Thanks a lot for reading our paper and for your insightful comments.
>
> 1. **Lack of intuitive understanding**: Thank you for your concern. To achieve a minimal-width prediction interval, it is crucial to accurately capture the shape of the interval. However, a single predictor may fail to do so due to its specific structural limitations. For example, the correct shape might be a polynomial function, but the predictor could output a linear function, resulting in model misspecification. By aggregating several predictors and considering a convex hull of these base predictors, we reduce model misspecification. This approach leads to an ensemble predictor that more accurately captures the interval's shape, resulting in a smaller width. The second step (shrinkage) is equivalent to conformal prediction, which maintains the coverage guarantee.
>
> 2. **Comparison with Hosen et al. (2014)**: Thank you for pointing out this interesting paper. We will include a comparison with the previous methods from Hosen et al. (2014) in our revised version.
> Hosen et al. (2014) focus on ensemble prediction intervals obtained using neural networks through ranking and weighted averaging. While they consider both coverage and width, they combine these metrics into a single optimization objective (coverage width criterion). In contrast, our method focuses on finding the interval with the smallest width among those that provide sufficient coverage.
> Additionally, Hosen et al. (2014) do not provide theoretical guarantees for coverage and width, nor do they address domain shift scenarios. Our work, on the other hand, emphasizes providing a theoretically robust methodology specifically designed to handle domain shift challenges.
>
>
> 3. **Limited experiments**: Thank you for raising this point. We have now conducted three additional real-data experiments to evaluate our method. We have added one more baseline, the weighted quantile conformal prediction method. Please see the global response for details on these experiments. Last but not least, we have also conducted a simulation experiment quantifying the robustness of our method against conformal prediction. Please see our response to Reviewer 6ToN for details of that experiment.
>
> 4. **What does $f \in \\mathcal{F}$ signify?**: Sorry for the typo, it should be $\min_{l, u}$.
>
> 5. **What does $m_0$ represent?**: $m_0$ is the conditional mean function, as defined in line 137. We will clarify it in our revision.
>
> 6. **Why not apply hinge to $\hat w(x)$?**: The purpose of using a hinge function is to ensure the convexity of the problem. Specifically, we replace the indicator function for $(Y - m_0(x))^2 - f(x)$ with a convex surrogate (hinge function) to facilitate the use of standard convex optimization techniques.
>
> 7. **Notation $\\mathcal{B}_{\mathcal{F}}$**: $\\mathcal{B}\_{\mathcal{F}}$ is defined to be an upper bound of $\sup_{f\in\mathcal{F}}\|f\|_{\infty}$ (as shown in line 181 in Theorem 3.2). We will make it more clear in our revision.
>
> 8. **Experiment on Algorithm 2**: We have conducted an experiment on the Airfoil data using our second algorithm, which requires estimating the optimal transport map. Please see our global response for the details.

---

> > ### Comment · Reviewer_EZSx · 2024-08-10
> >
> > Thank you for your detailed explanations. I appreciate the way you’ve clarified the intuitive understanding and your commitment to comparing it with previous approaches. However, in the quesiton 3, it states that "$\hat{\omega}$ may not be efficient in selecting informative source samples." Given this, would applying the hinge function to $(Y - m_0(x))^2 - f(x)$ help address this issue? Besides theoretical analysis convenience, are there any other benefits to this approach?

---

> > > ### Author Response · Authors · 2024-08-10
> > >
> > > Thanks for your further comments.
> > > In Step 1 (Shape estimation), the ideal optimization problem we aim to solve is given by equation (3.2) in our paper, which can be expressed as:
> > >
> > > $$
> > >   \\min\_{f \in \\mathcal{F}} \  \mathbb{E}\_{n, T}[f(X)]\, \ \ {\rm s.t.}\ \\mathbb{E}\_{n,S}[w_0(X)\\mathbf{1}_{\{f(X)<(Y-m_0(X))^2\}}]=0 \.\tag{1}
> > > $$
> > >
> > >
> > > If we had perfect knowledge of the source samples for which $ w_0 > 0 $, we could directly solve equation (3.2). However, in practice, $w_0$ is unknown, and we must estimate it using $ \hat{w} $. This introduces the possibility of estimation error, where $ w_0(X_i)=0 $ but $ \hat{w}(X_i)> 0 $ for some source samples $X_i $. Such errors lead to additional constraints—namely, requiring $ f(X_i)\geq(Y_i-m_0(X_i))^2 $ for all $X_i $ where $ \hat{w}(X_i)> 0 $—that are not necessary if $w_0(X_i)=0 $.
> > >
> > > To address this, instead of enforcing the right-hand side of the constraint in (1) to be exactly 0, we relax it to $ \epsilon $ to accommodate the estimation error in $\hat{w} $. This adjustment leads to the following optimization problem:
> > >
> > > $$
> > > \min_{f \in \\mathcal{F}} \ \ \\mathbb{E}_{n, T}[f(X)]\, \ \ \text{s.t. }\ \\mathbb{E}\_{n,S}\left[\hat{w}(X)\\mathbf{1}\_{\{f(X)<(Y-m_0(X))^2\}}\right]\leq \epsilon \.\tag{2}
> > > $$
> > >
> > > However, there is a caveat: the presence of the indicator function in (2) makes the constraint non-convex. To address this, we replace the indicator function with a convex surrogate, specifically the hinge function $ h_{\delta}\left((Y-m_0(X))^2-f(X)\right) $, which restores convexity to the problem.
> > >
> > > In summary, we start with equation (3.2) from our paper, substitute $w_0 $ with $ \hat{w} $, and then replace the indicator function with a hinge function to ensure the problem remains convex. Thus, the use of the hinge function is driven by computational considerations rather than merely for theoretical analysis.

---

> > > > ### Comment · Reviewer_EZSx · 2024-08-12
> > > >
> > > > Thanks for your detailed repsonse.  I have decided to increase the rating.

---

> > > > > ### Author Response · Authors · 2024-08-12
> > > > >
> > > > > Thank you very much for taking the time to read our rebuttal. We greatly appreciate your positive feedback and the decision to increase the score.

---

### Official Review · Reviewer_zRWP · 2024-07-10

**Soundness:** 2
**Presentation:** 3
**Contribution:** 2
**Rating:** 5
**Confidence:** 3

**Summary:**

The authors study the problem of how to construct prediction intervals on a target domain under both covariate shift and domain shift assumptions (i.e., the source and target domains are related either via a bounded density ratio, or a measure-preserving transformation), designed to ensure adequate coverage while minimizing interval width.

They provide theoretical guarantees, with finite sample bounds, regarding the prediction interval coverage and width.

They apply their method on the airfoil dataset, comparing the performance with weighted split conformal prediction.

**Strengths:**

- The paper is well-written overall, the authors definitely seem knowledgeable.

- The paper studies an important and interesting problem.

- The proposed method outperforms weighted split conformal prediction on one dataset.

**Weaknesses:**

- The experimental evaluation is limited to a single low-dimensional tabular dataset, with a synthetic/simulated distribution shift. It is not possible to determine if the proposed method actually would have significant real-world utility/impact.




Summary:
- Important problem and a well-written paper overall, but the experimental evaluation is too limited. I think this could be a solid paper, but I don't think it can be accepted in its current form.

**Questions:**

- Could you extend the experimental evaluation with more datasets and baseline methods?

- Why is the figure on page 9 missing a figure number and caption?

**Limitations:**

Yes.

---

> ### Author Rebuttal · Authors · 2024-08-06
>
> Thanks a lot for reading our paper and for your insightful comments.
>
> 1. **Extension of empirical evaluation**: Thank you for your comment, please see our global response for the details of the additional experiments.
>
> (i). We now have two more real-data experiments on our first method, Algorithm 1(which relies on the estimation of the density ratio of the source and the target covariates), one using the real estate valuation data and the other using the energy efficiency data. Furthermore, we have an additional experiment on our second method, Algorithm 2 (which relies on estimating the optimal transport map), using the Airfoil data.
>
> (ii). We have now added one more baseline, namely the weighted quantile-adjusted conformal method, i.e. we now compare our against both variance-adjusted weighted conformal and weighted quantile-adjusted conformal method.
>
> (iii). We have also conducted a simulation experiment to showcase the robustness of our algorithm in comparison to weighted conformal prediction. Kindly see our response to Reviewer 6ToN for the details of this experiment.
>
> 2. **Figure 9**: Thank you for pointing out this problem. We will take care of it in the revised version of the paper.

---

> > ### Comment · Reviewer_zRWP · 2024-08-08
> >
> > Thank you for your response.
> >
> > I have read the other reviews and all rebuttals.
> >
> > The other reviews are all borderline, but slightly more positive than mine. The authors add some new experiments, and respond well overall to the reviews.
> >
> > My main concern _"The experimental evaluation is limited to a single low-dimensional tabular dataset, with a synthetic/simulated distribution shift. It is not possible to determine if the proposed method actually would have significant real-world utility/impact"_ is not fully addressed. The evaluation is still limited to low-dimensional tabular datasets with synthetic/simulated distribution shifts, and I thus still find it difficult to know if the method actually would have significant real-world utility.
> >
> > - Could you evaluate on some dataset with a real-world distribution/domain shift?
> > - Could you evaluate on some non-tabular dataset, on an image-based dataset?
> >
> > - Also, is it not a fairly major concern that the coverage of your method falls below 0.95 in the "Airfoil data with Algorithm 2" experiment? Yes, the interval length is roughly half that of WVAC and WQC, but is this (big) improvement worth it if the method fails to achieve valid coverage?
> >
> > I am open to increasing my score at least to "5: Borderline accept", but would still like to see at least some more convincing experimental results.

---

> > > ### Author Response · Authors · 2024-08-10
> > >
> > > Thanks for your further comments.
> > >
> > > 1.**Could you evaluate on some dataset with a real-world distribution/domain shift?**:
> > >
> > > We applied our method to the ETDataset (ETT-small) from [1], which contains hourly-level data from two electricity transformers at two different stations, including measurements of load and oil temperature.
> > > Each data point consists of 8 features, including the date of the point, the predictive value "oil temperature", and 6 different types of external power load features.
> > > For our experiment, we used the data from one transformer during the period from July 1, 2016, to November 2, 2016, as our source data, and data from the same time period from the other transformer as our target data.
> > > As our source data and the target data are from different locations, we have a geographical covariate shift; see [1] for more details.
> > > Our results are as follows:
> > >
> > > **Table:** Experimental results for the ETDataset
> > >
> > > | Outcome   | Our Method | WVAC  | WQC   |
> > > |-----------|------------|-------|-------|
> > > | Coverage  | 0.976      | 0.982 | 0.842 |
> > > | Bandwidth | 41.525     | 57.9  | 54.981|
> > >
> > > [1] Haoyi Zhou, etc. Informer: Beyond Efficient Transformer for Long Sequence Time-Series Forecasting.
> > >
> > > 2. **Could you evaluate on some non-tabular dataset, on an image-based dataset?**
> > >
> > > The widely used non-tabular data in the field of distribution shift are various types of image data (e.g., MNIST-USPS digit data, Waterbird data etc.), which are generally used for classification tasks.
> > > However, our current method for constructing prediction intervals relies on the continuous response variable. Therefore, it cannot accommodate non-continuous responses. We acknowledge this limitation of our method and believe that this extension will be an interesting future research direction.
> > >
> > > 3. **Also, is it not a fairly major concern that the coverage of your method falls below 0.95 in the "Airfoil data with Algorithm 2" experiment? Yes, the interval length is roughly half that of WVAC and WQC, but is this (big) improvement worth it if the method fails to achieve valid coverage?**
> > >
> > > We believe the reason our method fails to achieve coverage is a small sample effect rather than the limitation of our methodology. As is evident from other real data experiments, our method achieves the nominal level of coverage. Therefore, our method *does not sacrifice coverage to reduce width*; our theoretical results indicate that asymptotically, our method achieves adequate coverage while having minimal width.

---

> > > > ### Comment · Reviewer_zRWP · 2024-08-11
> > > >
> > > > Thank you.
> > > >
> > > > For the image-based data, would it perhaps be possible to apply your method to one of the regression datasets from _How Reliable is Your Regression Model's Uncertainty Under Real-World Distribution Shifts?_ (TMLR, 2023)?

---

> > > > > ### Author Response · Authors · 2024-08-12
> > > > >
> > > > > Thank you very much for bringing this benchmark paper to our attention for evaluating uncertainty estimation methods. Since your last comment, we have made a concerted effort to apply our method to the data referenced in the paper. However, due to limited computational resources and the large scale of the data (both in terms of high dimensionality and the number of training and testing samples), we found it challenging to process the datasets within the short timeline available. Specifically, we encountered difficulties running the ``create\_datasets.py" script, embedding the images into vector space, and applying dimension reduction techniques to extract useful information from the images.
> > > > >
> > > > > Nevertheless, we will cite the paper you mentioned as a valuable source of non-tabular data with distribution shifts, which could be used to evaluate our methods in future work.
> > > > >
> > > > > We would also like to take this opportunity to highlight the key contributions of our paper:
> > > > >
> > > > > 1. **Methodology:** We propose two novel methods (Algorithm 1 and Algorithm 2) to address distribution shift challenges where no labels from the target domain are observed. Our approach is straightforward to implement and involves solving a convex optimization problem.
> > > > >
> > > > > 2. **Theoretical Guarantees:** We establish rigorous theoretical guarantees, including finite-sample concentration bounds, to demonstrate that our method achieves adequate coverage with a small width.
> > > > >
> > > > > 3. **Experiments:** We evaluate Algorithm 1 on three real-world datasets: the Airfoil data, Real Estate Dataset, and Energy Efficiency data. Additionally, we assess Algorithm 2 on the Airfoil data. We further demonstrate the utility of our method by applying it to the ETDataset, which presents a non-synthetic geographical covariate shift. Finally, we highlight the advantages of our method compared to conformal prediction through robustness checks.
> > > > >
> > > > > We sincerely hope that our paper's contributions are properly recognized and that our responses have addressed most of your concerns. Thank you again for your valuable feedback. Please do not hesitate to let us know if there are any other concerns that might be preventing you from adjusting your score.

---

> > > > > > ### Comment · Reviewer_zRWP · 2024-08-13
> > > > > >
> > > > > > Thank you again for the response.
> > > > > >
> > > > > > While I realize that you don't have a lot of time to implement new experiments, I do still find it quite problematic that it hasn't been demonstrated that the proposed method actually can be applied to image-based datasets. My main concern _**"It is not possible to determine if the proposed method actually would have significant real-world utility/impact"**_ therefore mostly remains. Also, this doesn't quite seem to align with your statement _"Our approach is straightforward to implement"_ from above.
> > > > > >
> > > > > > You have however provided a detailed rebuttal, I can appreciate that there are some other contributions of the paper, and the other reviewers are now all positive overall. I have raised my score to "4: Borderline reject" for now.

---

> > > > > > > ### Author Response · Authors · 2024-08-13
> > > > > > >
> > > > > > > Thank you for your feedback and for the increased score. We appreciate your concerns and would like to address them in the following:
> > > > > > >
> > > > > > > 1. **Real-World Utility**: While we were unable to demonstrate the utility of our method on image-based data within the limited time available (notably, the AssetWealth dataset you referenced is 13GB), it's important to highlight that numerous real-world applications extend beyond image-based tasks. Examples include price prediction in financial markets, weather forecasting, and energy consumption analysis, among others. Given the vast array of real-world tasks, it's impractical to test our method on all possible datasets. However, we've showcased its utility through experiments on the Airfoil dataset (sound pressure prediction), Real Estate Dataset (house price prediction), Energy Efficiency data (heating load prediction), and ETDataset (oil temperature prediction). These experiments consistently demonstrate the advantages of our method over WVAC and WQC, suggesting its significant potential impact across diverse real-world scenarios. Additionally, ***the primary contributions*** of our paper lie in its methodological and theoretical advancements, particularly the finite sample coverage and width guarantees, as previously mentioned.
> > > > > > >
> > > > > > > 2. **Implementation Ease**: Our approach is straightforward to implement in that it primarily involves solving convex or linear programming problems. However, processing large datasets, especially image-based ones, can be inherently time-consuming due to steps like embedding images into vector spaces and performing dimensionality reduction. Once these preprocessing steps are completed and we obtain vector representations of the images, our method can be implemented efficiently and executed rapidly.
> > > > > > >
> > > > > > > We genuinely value your insights and believe that our method holds promise for various applications.
> > > > > > > We would greatly appreciate it if you could consider further raising your score above the acceptance threshold if you find our clarifications satisfactory.

---

> ### Comment · Reviewer_zRWP · 2024-08-13
>
> _(I will quickly write a reply now, to try to give you some time to respond before the deadline, but I will also think more carefully about this later before making my final recommendation. I will still consider raising my score)_
>
> ***
> ***
>
>
> > (notably, the AssetWealth dataset you referenced is 13GB)
> - There are 8 different datasets in the TMLR paper i pointed to, I thought that maybe it could be easy to apply to at least one of those?
>
> > ...suggesting its significant potential impact across diverse real-world scenarios
> - The method has only been applied to a single dataset with a real-world distribution shift though, right? Are there some other tabular datasets with realistic distribution shifts?
>
> > These experiments consistently demonstrate the advantages of our method over WVAC and WQC
> - Expect for "Airfoil data with Algorithm 2" where the coverage drops below 0.95? Also, is this the only evaluation of Algorithm 2? Why? (could perhaps be made more clear in general what the two different algorithms should be utilized for, how do I know which one to use in a real-world application?)
>
> > ...performing dimensionality reduction. Once these preprocessing steps are completed and we obtain vector representations of the images, our method can be implemented efficiently and executed rapidly
> - Why is dimensionality reduction a required step? Isn't this a quite significant limitation of your method then, which would limit the real-world utility also for non-image data? No dataset in the experiments has more than 8(?) features, what's the maximum number your method can handle?

---

> > ### Author Response · Authors · 2024-08-13
> >
> > Thanks for your further comments.
> >
> > 1. **The method has only been applied to a single dataset with a real-world distribution shift though, right? Are there some other tabular datasets with realistic distribution shifts?**
> >
> > Following your concern, we managed to run an additional experiment within this short timeframe. The new dataset is the *Appliances Energy Prediction Dataset*, which is freely available from the UCI repository.
> > This dataset is a time series data with 28 covariates and one response variable.
> > We used data from 2016-01-11 to 2016-02-15 (5000 samples) as our training set and data from 2016-05-13 to 2016-05-27 (2000 samples) as our testing set. Since the source and target data are from different time periods, this experiment involves a non-synthetic real-world time shift. The results based on our Algorithm 2 are presented below:
> >
> > **Table:** Experimental results for the Appliances Energy Prediction Dataset
> >
> > | Outcome   | Our Method | WVAC     | WQC     |
> > |-----------|------------|----------|---------|
> > | Coverage  | 0.95       | 1.00     | 1.00    |
> > | Bandwidth | 461.69     | 6809.87  | 2032.12 |
> >
> >
> > 2. **Why is dimensionality reduction a required step?**
> >
> > We would like to clarify that our experiment involves two main steps: first, training the component functions, and second, efficiently aggregating these components. Our methodology specifically focuses on the aggregation step, where we combine pre-trained models to obtain a prediction interval with adequate coverage and minimal width. The key point is that our method assumes these component functions are already trained, and our contribution lies in the efficient aggregation of these components. When we say our method is easy to implement, we are referring to this aggregation step, which is independent of the data's dimensionality.
> >
> > In many real-world applications, these components are typically pre-trained and provided to us. However, we acknowledge that if the pre-trained components are not available, they must first be trained from the data. This training process can indeed be time-consuming, especially as the models' complexity increases. The dimension reduction step may be required for training these component functions, but is **not required** for aggregation.
> >
> > To summarize, when we refer to *our method*, we mean the aggregation (shape estimation) and shrinkage steps, which are straightforward to implement and do not depend on the data’s dimensionality. The more challenging aspect is the initial training of the components if they are not already available.
> >
> >
> > Our method can indeed handle large datasets, with the main consideration being the time required to compute the component functions. To illustrate, we conducted an additional experiment on Appliances Energy Prediction Datase, using data with 28 covariates. It's important to note that the aggregation step involves combining multiple component functions, each of which is univariate. Therefore, the dimension involved in the aggregation step is the number of component functions, **not the dimension of the data**; the data's dimensionality only affects the training of the component functions.
> >
> > 3. **Expect for "Airfoil data with Algorithm 2" where the coverage drops below 0.95? Also, is this the only evaluation of Algorithm 2? Why?**
> >
> > As we mentioned, the drop in coverage below 0.95 is due to the small sample effect rather than a limitation of our methodology. We evaluated Algorithm 2 on both the Airfoil data and the ETDataset. The choice between Algorithm 1 and Algorithm 2 depends on prior knowledge about the type of distribution shift. If the shift is due to covariate changes, and the support of the test covariates lies within the support of the source covariates, then Algorithm 1 is appropriate. However, Algorithm 2 is more suitable when the shift is caused by transformations (e.g., shifts, rotations). If the support of the source and target covariates is likely to be (almost) disjoint, density ratio estimation will be poor, making Algorithm 2 the better option in such cases.
> >
> > 4. **There are 8 different datasets in the TMLR paper I pointed to, I thought that maybe it could be easy to apply to at least one of those?**
> >
> > All eight datasets referenced in the TMLR paper are large-scale datasets. Even the smallest among them, SkinLesionPixels, is still 2GB. Additionally, accessing some of these datasets requires account registration, which further complicates the process.

---

> > > ### Comment · Reviewer_zRWP · 2024-08-14
> > >
> > > Good clarifications, thank you.
> > >
> > > And thanks for the detailed responses throughout, I appreciate the effort put into this.
> > >
> > > I still really would have liked to see the proposed method being applied to non-tabular datasets, but I'm now leaning towards accept either way. I will raise my score to "5: Borderline accept".

---

> > > > ### Author Response · Authors · 2024-08-14
> > > > **Official comment by Authors**
> > > >
> > > > Thank you very much for your suggestions and responses throughout the review session! We believe this has truly helped our paper, and we will add all the additional details to the revised version of the manuscript.

---

### Official Review · Reviewer_88pD · 2024-07-10

**Soundness:** 3
**Presentation:** 2
**Contribution:** 3
**Rating:** 6
**Confidence:** 4

**Summary:**

Building on the work of Fan et al. (2023), this paper addresses the challenge of computing prediction intervals in an unsupervised domain adaptation setting, where labeled samples are available from a related source domain, and unlabeled covariates are available for the target domain. The primary objective is to generate prediction intervals for the target domain that provide both an asymptotic coverage guarantee and minimal width. The authors also explore the aggregation of prediction intervals on the target domain under both covariate shift and domain shift scenarios. Theoretical guarantees, including finite sample concentration bounds on approximation errors and coverage guarantees, are presented. The proposed methods are illustrated and compared with the weighted split conformal prediction method using a real-world dataset.

**Strengths:**

- The paper tackles the important and challenging problem of uncertainty quantification in the unsupervised domain adaptation setting.

- The authors propose algorithms to generate prediction intervals with both coverage and minimal width guarantees.

- Experimental results show that the proposed method generates prediction intervals with comparable coverage but smaller average width compared to an existing method.

**Weaknesses:**

- The paper assumes familiarity with Fan et al. (2023), making it difficult for readers who have not read that work to understand some notations and concepts. Section 2.1 on problem formulation is particularly hard to read without prior knowledge.
	- There is a lack of flow in the paper, with some explanations and definitions missing or unclear. For instance, terms like "valid prediction interval", and expressions such as U(x) and l(x) are not clearly explained.
	- The authors write "This translates into solving the following optimization problem:" without any explanations.
	- Another example is Line 234, "adding a slight delta to ensure coverage even when F is complex.". This is not clear. It is discussed in Fan et al (2023) in (2.4).
	- Even if simple, I suggest adding important details about different expressions in the appendix to help the reader.


- The title mentions "optimal aggregation," but model aggregation is not the central focus of the paper. It is first discussed in Remark 4.2 and briefly mentioned in line 272.

- The paper should highlight the similarities and differences with Fan et al. (2023) in terms of contributions, both theoretical and empirical.

Also, the authors use different quantile levels for estimators compared to Fan et al. (2023) without explaining the reason for these differences.  In Fan et al (2023), the following quantile levels are considered: 0.05, 0.35, 0.65, and 0.95, while the authors use 0.85, 0.95, 0.9, and 0.9.


- The statement that "the shape of the prediction band does not change much if we change the level of coverage" lacks clarity on scenarios where this might not be true, such as multimodal distributions (?).


- The discussion on lambda <=1 is confusing and needs improvement. The paper inconsistently explains the shrinkage factor and its implications. For example, the authors write "Therefore, it is not immediate whether the shrinkage factor  is smaller than 1, i.e., whether we are indeed shrinking the confidence interval (lambda > 1 is undesirable, as it will widen finit , increasing the width of the prediction band)." and "Although ideally lambda <= 1, it is not immediately guaranteed as we use separate data  for shrinking.". The authors also discuss it in Lemma 3.3. and Lemma 4.4.


- The term "shape of the prediction band" with prediction intervals can be confusing as it can have other meanings (see functional data). I suggest the authors clarify this in line 147.


- The paper ignores a significant amount of research on prediction interval estimation. I suggest discussing important related work.

- The limitations of the paper are not discussed. For example, focusing solely on intervals might have implications under multimodal distributions, where High-Density Regions (HDR) would be the smallest regions.

- The relationship between the proposed approach and conformal prediction is not clearly explained. The paper should discuss the guarantees provided by the proposed approach compared to conformal prediction, including the relevant theoretical guarantees and calibration aspects.

For example, the weighted conformal approach also splits the data and estimates a density ratio.  What about the (weaker?) guarantees you provide? The following paper could also be useful: "On the expected size of conformal prediction sets".

Also, in conformal prediction, in addition to finite-sample marginal guarantees, the authors often provide asymptotic conditional guarantees. What can you say about this with your approach?


- The comparison with the conformal approach in the experiments might not be fair, as conformal prediction does not optimize for the smallest width. Additionally, reporting the interval score, which is a proper scoring rule for intervals, would be useful for evaluating the methods.


- Typos and clarifications:
	- line 137: "remains same."
	- line 154: "via by solving"
	- line 453 "a.s. on source". On the source domain?
	- line 470, "we with have"
	- It is not "Leduox-Talagrand" but Ledoux-Talagrand".
        - line 481, a minus sign is missing
	- Line 490 to 492. It is not clear why tilde tau <= alpha.
	- Line 182, what does \mathcal{F} - f* mean?
	- About Theorem 3.4, we expect the performance (or convergence rates) to also depend on alpha. Where does this appear in the theoretical guarantees?

**Questions:**

See weaknesses.

**Limitations:**

The limitations are not discussed in the paper.

---

> ### Author Rebuttal · Authors · 2024-08-06
>
> Thanks a lot for reading our paper and for your insightful comments.
>
> 1. **The paper assumes familiarity with Fan et al. (2023)** Thank you for raising this concern! We will carefully proofread the revised version of the manuscript according to your suggestions.
>
> 2. **Mention optimal aggregation** Thank you for pointing this out. Indeed, our current approach presents the general problem and then casts prediction interval aggregation as a special case. We will highlight the aggregation perspective more in the revised version of the manuscript.
>
>
> 3. **Similarities and dissimilarities to Fan et al. (2023)** Please see our global response. We selected the quantile levels of 0.85, 0.95, 0.9, and 0.9 because they are close to the desired coverage and we believe they effectively capture the correct shape of the interval. It is important to note that our method is not restricted to these specific quantiles; we can also apply the quantile levels used in Fan (2023) to our approach if needed.
>
>
> 4. **Change of shape of prediction band** Thank you for your feedback. We will clarify this in line 147 in the revision. Specifically, when we refer to the "the shape of the prediction band does not change much", we mean that the form of the minimal average width prediction band covering the data at a certain coverage level should not change much if we slightly change the coverage level, i.e., the form is stable with respect to the perturbation of the coverage level. For instance, if the optimal prediction band that covers $98\\%$ of the data is quadratic in $X$, then the minimal-width interval that covers $95\\%$ of the data is also a quadratic function of $X$, rather than abruptly changing to a linear function. Under this consistency assumption, a constant shrinkage factor $\lambda$ in the second step (i.e., (3.3) and (4.2)) is sufficient to adjust from $100\\%$ to $95\\%$ coverage. However, if this is not the case, we can modify our method to allow the shrinkage level $\lambda$ to depend on the input $X$, ensuring more accurate adaptation to the data distribution.
> For the multimodal distributions, we acknowledge that the optimal region is not necessarily a prediction band (i.e., $Y|X=x$ is not necessarily a single interval) and will include this in the limitation of the work.
>
> 5. **Discussion on $\lambda \le 1$** Thank you for the comments. We will clarify this discussion in the revised version of the paper. In the first step, we typically choose $\epsilon$ to be small; therefore, the average width of $\hat f_{\rm init}$ can be larger than necessary, and coverage is also typically larger than $95\\%$. Therefore, in the second step, we aim to shrink the interval (i.e., ensure $\lambda \leq 1$). However, it is not immediate that $\hat \lambda$, obtained by solving equation (3.3) is indeed $\le 1$. To ensure this, we prove in Lemma 3.3 and Lemma 3.4 that with high probability, we indeed have $\hat \lambda(\alpha) \le 1$.
>
>
> 6. **Research on prediction interval estimation**
> Thank you for pointing this out. In our revision, we will add a detailed related work section that covers significant research on prediction interval estimation, including methods applicable to both shifted and non-shifted data. We will discuss approaches such as conformal prediction, quantile regression, and RKHS-based methods, highlighting their contributions and relevance to our work.
>
>
> 7. **Limitations of the paper are not discussed** Thank you very much for highlighting this limitation! It is indeed true that if the conditional distribution of $Y$ given $X$ is multimodal, the prediction interval may not be the best thing to do. We will mention this in the limitation.
>
> 8. **Comparison with conformal**
>
> (1) *Methodology relation and difference*: Step 2 of Algorithm 1, i.e., shrinkage (equation (3.3) and (4.2)) in our method is equivalent to (weighted) conformal prediction, using the score function $(y - \hat{m}(x))/\hat{f}(x)$, where $\hat{f}$ is obtained from the first step of our method. By efficiently aggregating various predictors, our $\hat{f}$ accurately captures the shape of the interval, resulting in a smaller bandwidth. In contrast, variance-adjusted conformal prediction selects $f$ as the variance function, which may not necessarily capture an optimal shape for a small bandwidth. Additionally, our method explicitly aims to minimize the bandwidth in the first step. In contrast, conformal prediction does not explicitly minimize bandwidth; it implicitly addresses this by setting the cutoff at the $1-\alpha$ quantile of the score function.
>
> (2)*Robustness*: Our method is more robust compared to variance-adjusted conformal prediction. Please see our response to Reviewer 6ToN for the details of our experiment regarding robustness.
>
> (3)*Theoretical guarantees*: We provide finite sample guarantees for both the width and coverage of our prediction intervals. In contrast, while typical conformal literature offers finite-sample marginal or asymptotic conditional guarantees for coverage, they often do not include finite-sample guarantees for width, as our paper does.
> For distribution shift [1] provides coverage guarantee assuming the density ratio $w(\cdot)$ is known, whereas our Theorem 3.4 explicitly highlights the effect of the estimation error of $w(\cdot)$.
> We acknowledge that our method is particularly focused on addressing marginal coverage, and therefore, its pointwise guarantee is not immediately clear. We will include this as a limitation in our discussion and consider it a topic for future research.
>
> [1] Tibshirani, R., et al., Conformal prediction under covariate shift.
>
> 9. **Typo** Thank you very much for carefully reading our paper and pointing out the typos! We will take care of it in the revised version of the manuscript.

---

> > ### Comment · Reviewer_88pD · 2024-08-10
> >
> > Thank you for answering my questions and concerns. I have raised my score to 6 (weak accept).

---

> > > ### Author Response · Authors · 2024-08-10
> > >
> > > Thank you very much for both your valuable reviews and the increased score. We will carefully revise our manuscript based on your suggestions.

---

### Official Review · Reviewer_6ToN · 2024-07-12

**Soundness:** 3
**Presentation:** 3
**Contribution:** 2
**Rating:** 5
**Confidence:** 4

**Summary:**

This work proposes a method to construct prediction intervals in distribution shift problems, where unlabeled data from the target domain is available. The method is inspired by Fan et al (2023) in the i.i.d. setting and assumes we can estimate a transform (defined by reweighting or a transport map) that maps the joint distribution from the source domain to the target.

**Strengths:**

- The problem studied is relevant.
- The method appears reasonable and sound.

**Weaknesses:**

- My main concern is about the novelty of this work: given the assumed condition on domain shift (namely a good approximation to the reweighting or transport map is readily available) it is a natural application of the method in Fan et al (2023).  The proof will require some additional effort but given the assumptions I do not expect them to be technically challenging.

- While the paper employed assumptions on domain shift are common in literature, I am uncertain about their relevance, especially in high-dimensional problems.  It is well-known that the bounded density ratio assumption in Section 3 is unrealistic in high dimensions. And while I have come across past works on the transport map assumption, I do not know of realistic examples where the assumption actually holds so that the coverage guarantees established in this work could be relevant: it is much easier to think of scenarios where the assumption does not necessarily hold, but heuristic methods based on this assumption achieve better predictive performance; the latter is more often the goal in previous works.

- Finally, I have doubts about the novelty/utility of Fan et al (2023), which seems quite similar to conformal prediction.  It could be alternatively summarized as applying quantile estimation to the "conformity score" of $(y - \hat m(x)) / f(x)$, and compared with conformal prediction the main difference is the lack of sample splitting (in split conformal) or leave-one-out operations (as in full conformal).  I wonder if the removal of sample splitting is really a wise choice: the coverage guarantees in Fan et al (2023) and this work relies on the Rademacher complexity of the function class for $f$.  If the function class is defined by general ML models the guarantees would be rather unreliable in high-dimensional problems, in sharp contrast to the distribution-free guarantees in (split/full) conformal prediction.  Otherwise, we would need further sample splitting to construct a finite set of candidates for $f$, in which case we could just do split conformal prediction.

**Questions:**

Given the above questions on the theoretical motivation, novelty or significance, I believe there should be extensive empirical evaluation for the proposed method.

In light of question 3 above, I also wonder if the observed performance difference between the proposed method and conformal prediction could be attributable to the choice of different conformity scores; how does the method compare to an implementation of split conformal prediction using the same $(y - \hat m(x)) / f(x)$ as the conformity socre?

**Limitations:**

Yes.

---

> ### Author Rebuttal · Authors · 2024-08-06
>
> Thanks a lot for reading our paper and for your insightful comments. We answer your questions and concerns in the following.
>
> 1. **Novelty compared to Fan et al. (2023)**:  Please see our global response.
>
> 2. **Assumptions on the covariate shift**:  Covariate shift is a common assumption, even for high-dimensional data, e.g., see [1] or [2]. The impact of domain shift in our analysis is primarily through the estimation error of the density ratio and the optimal transport map. We acknowledge that high-dimensional settings can pose challenges, such as the curse of dimensionality. However, these issues can often be mitigated by incorporating more structure; without any structural assumption, we do not expect to gain any theoretical guarantee under covariate shift in high-dimension. An example of a structured distribution shift is the *exponential tilt* assumption (the one we used in our experiments),  or that only a subset of variables undergoes the shift (sparsity type assumption).
> Similarly, we can assume some structure in the optimal transport map to avoid the curse of dimensionality. While our analysis is quite general, we believe that with specific structural assumptions, our method and analysis can be appropriately adjusted to incorporate these structures. Therefore, we agree that we need some structural assumptions in high-dimension, but without any assumption, the curse of dimensionality is unavoidable. Once we have that, our method is applicable to obtain a valid prediction band with a small width.
>
> [1] Tripuraneni, N., et al., Covariate shift in high-dimensional random feature regression.
>
> [2] Tripuraneni, N., et al., Overparameterization improves robustness to covariate shift in high dimensions.
>
> 3. **Similarity and dissimilarity with conformal prediction**: We agree with you that there is a similarity between our method and the conformal prediction in the second stage, i.e., the shrinkage (i.e., (3.3) and (4.2)), which is equivalent to (weighted) conformal prediction as both of these methods aim to derive weighted quantile of the sample score $s(x, y) = (y - \hat{m}(x))/\hat{f}(x)$. However, the key difference lies in the construction of $\hat f$; by efficiently aggregating various predictors, $\hat f$ accurately captures the shape of the interval, resulting in a smaller bandwidth.
>
> # Robustness check
> Another advantage of our method is its robustness. Our method aggregates various predictor intervals, including an estimator for the conditional variance function (Estimator 5). In contrast, the weighted variance-adjusted conformal (WVAC) prediction interval [3] heavily relies on accurately estimating this conditional variance, while conditional quantile regression relies on accurate estimation of the conditional quantile function. We did a small robustness check using the following model:
> \begin{align*}
>     & X \sim{\sf Unif}([-1, 1]), \ \ \xi \sim{\sf Unif}([-1, 1]),\ \  X,\xi \text{ independent}, \\
>     & Y = \sqrt{1 + 25 X^4} \xi .
> \end{align*}
>
> We estimated the conditional variance using a random forest with varying depths \{3, 5, 7, 15\}. We generated n = 2500 samples, keeping $75\\%$ as source data and resampling the remaining 25\% with weighted samples proportional to $w(x) \propto (1 + \exp{(-2x)})^{-1}$. As depth increases, overfitting leads to poor out-of-sample variance predictions. The following table is the result of 100 Monte Carlo experiments. The number inside the parenthesis is the median of coverage over these Monte Carlo iterations.
>
> | Max depth | Avg. width--Our | Avg. width--WVAC |
> |-----------|------------------|------------------|
> | 3         | 2.07 (0.975)    | 3.08 (0.9712)    |
> | 5         | 2.07 (0.95)      | 3.28 (0.9664)    |
> | 7         | 2.068 (0.94)    | 3.33 (0.97)      |
> | 15       | 2.08 (0.97)      | 5.00 (0.97)      |
>
> The above table implies our method is more robust to the misspecification of some model components and remains stable.
>
> In a nutshell,  our method has a similar coverage guarantee as the conformal prediction, which does not require many assumptions; our coverage guarantee (Theorem 3.4) requires a reasonable estimation of $w(\cdot)$, which is necessary for both our method and that of [3]; [3] even assumed $w(\cdot)$ is known, whereas our bounds quantify the estimation error of $w(\cdot)$. When we require a guarantee on the width, we require some assumptions on the predictor class.
>
> [3] Tibshirani, R., et al., Conformal prediction under covariate shift.
>
> 4. **Extensive empirical evaluation** Please see our global response.

---

> > ### Comment · Reviewer_6ToN · 2024-08-13
> >
> > Thank you for your response. I appreciate the clarifications regarding the theoretical contributions and the new experiments. I will update my score accordingly.

---

> > > ### Author Response · Authors · 2024-08-13
> > >
> > > Thank you very much for reading our rebuttal and for increasing the score. We will put the new experiments in our revision.

---

> ### Author Response · Authors · 2024-08-12
>
> As the rebuttal period is coming to a close, we wanted to check if there are any remaining concerns that might be preventing you from adjusting your score. If there is any additional clarification we can provide, please let us know. We would be more than happy to address any questions you might have. Thank you again for your time and effort in reviewing our paper.

---

### Author Rebuttal · Authors · 2024-08-06

# Global response
We thank all the reviewers for their insightful comments. We address a few concerns that were raised by multiple reviewers.
1. **Comparison with Fan et al. (2023):** The key contribution of our paper lies in adapting the methodology from Fan (2023) (which only addresses no-shift scenarios), to tackle domain shift challenges.
We deal with unsupervised domain adaptation, i.e., i) the distribution of the covariates of the target domain is different from the source domain, and ii) we do not observe any label from the target domain. Therefore, we are required to change the methodology;
to adapt to these challenges, the density ratio or the optimal transport map between the covariates must be estimated. Furthermore, as pointed out in Section 3, the shift may cause the optimization problem non-convex, for which we need to introduce a convex surrogate (e.g., the hinge function). As a consequence, we have various additional theoretical challenges to establish that our method indeed produces a prediction band on the target domain with adequate coverage and minimal width. For example, our theory highlights the precise effect of the estimation error of the density ratio or the optimal transport map in the bound on the coverage and the width. Furthermore, when we replace the indicator function with its convex surrogate hinge function (see (3.5)), we introduce two additional parameters $(\epsilon, \delta)$, effectively measuring the closeness between the original NP-hard problem (involving indicator function) and its convex surrogate. Our theoretical bounds also highlight the effect of these hyperparameters in our finite sample bounds.

2. **Additional Experiments:** We conducted three more real-data experiments.
In the following, WVAC is weighted variance adjusted conformal prediction interval, where the score function is $s(x, y) = |y - \hat m(x)|/\hat \sigma(x)$ with $\hat m$ and $\sigma$ being the estimate of conditional mean and variance function respectively. WQC is the weighted quantile conformal prediction interval with $s(x, y) = \max \\{\hat q_{\alpha/2}(x) - y, y - \hat q_{(1-\alpha)/2}(x)\\}$, where $\hat q$ is the estimated quantile function.

(a). **Real Estate Dataset** The Real Estate Valuation dataset consists of 414 instances. The goal is to predict the house price per unit area based on the 6 other features. The data is available at the UCI ML repository. The construction of shifted data (with $\beta = (-1, 0, -1, 0, 1, 1)$) and implementation procedure are the same as our paper. The following table and Figure 1 in PDF present the results over 200 Monte Carlo iterations. It is evident from the table that our method produced a small average width in comparison to the other methods while maintaining the coverage guarantee.

| Outcome                                         | Our Method | WVAC    | WQC     |
|-------------------------------------------------|------------|---------|---------|
| **Coverage (Median)**                           | 0.98       | 0.962   | 0.971   |
| **Coverage (IQR)**                              | 0.058      | 0.048   | 0.048   |
| **Bandwidth (Median)**                          | 36.889     | 46.392  | 46.858  |
| **Bandwidth (IQR)**                             | 15.001     | 19.027  | 14.189  |
| **Bandwidth (Median for Coverage > 95%)**       | 40.774     | 50.703  | 51.031  |




(b). **Energy efficiency data** The goal of the Energy Efficiency dataset is to predict the heating load based on 8 other covariates.
The construction of shifted data (with $\beta = (-1, 0, 1, 0, -1, 0, 0, -1)$) and implementation procedure are the same as our paper. This data is also available at the UCI ML repository. The following table and Figure 2 in PDF present the results over 200 Monte Carlo iterations. It shows that our method produced a smaller bandwidth than WVAC. While WQC has a smaller median bandwidth, it sacrifices coverage. The last row indicates that for experiments with coverage $\geq 95\\%$, WQC's median average width is significantly larger than ours. Thus, whenever WQC provides adequate coverage, its bandwidth is much larger than ours.
| Outcome                                         | Our Method | WVAC    | WQC     |
|-------------------------------------------------|------------|---------|---------|
| **Coverage (Median)**                           | 0.995      | 0.969   | 0.973   |
| **Coverage (IQR)**                              | 0.047      | 0.036   | 0.05    |
| **Bandwidth (Median)**                          | 4.332      | 5.045   | 2.842   |
| **Bandwidth (IQR)**                             | 1.358      | 3.269   | 2.551   |
| **Bandwidth (Median for Coverage > 95%)**       | 4.373      | 5.681   | 4.94    |

(c). **Airfoil data with Algorithm 2** We implement our second method (based on OT) using the airfoil data. Here, we shift 25% of the data by a linear transformation: $x \mapsto Ax + b$, where $A$ = diagonal(1.5, 1.2, 1.6, 2, 1.8) and $b = (1, 0, 0, 1, 0)$. Our method produces a small width in comparison to other methods. See Figure 3 in PDF and the following table.

| Outcome                                         | Our Method | Our Method (without OT) | WVAC    | WQC     |
|-------------------------------------------------|------------|-------------------------|---------|---------|
| **Coverage (Median)**                           | 0.928      | 0.749                   | 0.984   | 0.952   |
| **Coverage (IQR)**                              | 0.035      | 0.22                    | 0.024   | 0.077   |
| **Bandwidth (Median)**                          | 15.075     | 18.512                  | 36.298  | 32.143  |
| **Bandwidth (IQR)**                             | 1.638      | 3.089                   | 10.619  | 8.364   |
| **Bandwidth (Median for Coverage > 95%)**       | 16.429     | 25.268                  | 37.783  | 36.433  |

(d).**For the robustness experiment, please see our response to Reviewer 6ToN.**

---

### Decision · Program_Chairs · 2024-09-25

**Decision:**

Accept (poster)

**Comment:**

The authors propose a method to optimally combine prediction intervals under domain shift via convex optimization.

Reviewers had initial concerns but they were judiciously clarified by the authors.

I recommend accepting this paper.